# Strong confinement of active microalgae leads to inversion of vortex flow and enhanced mixing

**Debasmita Mondal[1], Ameya G Prabhune[1†], Sriram Ramaswamy[2], Prerna Sharma[1*]**

[1]Department of Physics, Indian Institute of Science, Bangalore, India; [2]Centre for Condensed Matter Theory, Department of Physics, Indian Institute of Science, Bangalore, India

**Abstract** Microorganisms swimming through viscous fluids imprint their propulsion mechanisms in the flow fields they generate. Extreme confinement of these swimmers between rigid boundaries often arises in natural and technological contexts, yet measurements of their mechanics in this regime are absent. Here, we show that strongly confining the microalga *Chlamydomonas* between two parallel plates not only inhibits its motility through contact friction with the walls but also leads, for purely mechanical reasons, to inversion of the surrounding vortex flows. Insights from the experiment lead to a simplified theoretical description of flow fields based on a quasi-2D Brinkman approximation to the Stokes equation rather than the usual method of images. We argue that this vortex flow inversion provides the advantage of enhanced fluid mixing despite higher friction. Overall, our results offer a comprehensive framework for analyzing the collective flows of strongly confined swimmers.

**\*For correspondence:**
prerna@iisc.ac.in

**Present address:** †Department of Physics, University of Colorado, Boulder, United States

**Competing interest:** The authors declare that no competing interests exist.

## Editor's evaluation

The manuscript focusses on changes in the flow fields generated by swimming microorganisms as a consequence of them being squeezed within a very narrow gap. The resulting friction with the cell body is such that the flows are dominated by the propelling appendages only, and the authors show in this regime the surprising result that there can be an enhanced nutrient flux to the microorganism.

## Introduction

Fluid friction governs the functional and mechanical responses of microorganisms which operate at low Reynolds number. They have exploited this friction and developed drag-based propulsive strategies to swim through viscous fluids (*Lauga and Powers, 2009*; *Pedley and Kessler, 1992*). Naturally, many studies have elucidated aspects of the motility and flow fields of microswimmers in a variety of settings that mimic their natural habitats (*Elgeti et al., 2015*; *Bechinger et al., 2016*; *Denissenko et al., 2012*; *Bhattacharjee and Datta, 2019*). The self-propulsion of microbes in crowded and strongly confined environments is one such setting, encountered very commonly in the natural world as well as in controlled laboratory experiments. Examples include microbial biofilms, bacteria- and algae-laden porous rocks or soil (*Qin et al., 2020*; *Hoh et al., 2016*; *Foissner, 1998*; *Bhattacharjee and Datta, 2019*); parasitic infections in crowded blood streams and tissues (*Heddergott et al., 2012*); and biomechanics experiments using thin films and microfluidic channels (*Durham et al., 2009*; *Denissenko et al., 2012*; *Jeanneret et al., 2019*; *Ostapenko et al., 2018*; *Kurtuldu et al., 2011*). Confined microswimmers are also fundamentally interesting as active suspensions (*Brotto et al., 2013*; *Maitra*

*et al., 2020*) and there are efforts to mimic these by chemical and mechanical means for applications in nano- and microtechnologies (*Duan et al., 2015*; *Temel and Yesilyurt, 2015*).

The mechanical interaction of microswimmers with confining boundaries alters their motility and flow fields (*Lauga and Powers, 2009*; *Brotto et al., 2013*; *Mathijssen et al., 2016*), leading to emergent self-organization in cell–cell coordination (*Riedel et al., 2005*; *Petroff et al., 2015*), spatial distribution of cells (*Tsang and Kanso, 2016*; *Rothschild, 1963*), and ecological aspects such as energy expenditure, nutrient uptake, fluid mixing, transport, and sensing (*Lambert et al., 2013*; *Pushkin and Yeomans, 2014*). It is expected that steric interactions will dominate with increasing confinement at the swimmer–wall interface and that hydrodynamic screening by the confining wall will lead to recirculating flow patterns or vortices (*Persat et al., 2015*; *Mathijssen et al., 2016*).

Among the abundant diversity of microswimmers, the unicellular and biflagellated algae *Chlamydomonas reinhardtii* (CR), with body diameter $D \approx 10\,\mu m$, are a versatile model system, widely used for understanding cellular processes such as carbon fixation, DNA repair and damage, phototaxis, ciliary beating (*Sasso et al., 2018*; *Brumley et al., 2015*; *Choudhary et al., 2019*; *Mondal et al., 2020*), and physical phenomena of biological fluid dynamics (*Goldstein, 2015*; *Brennen and Winet, 1977*; *Rafaï et al., 2010*). They are considered next-generation resources for wastewater remediation and synthesis of biofuel, biocatalysts, and pharmaceuticals (*Hoh et al., 2016*; *Khan et al., 2018*). Recently, extreme confinement between two hard walls has been exploited to induce stress memory in CR cells towards enhanced biomass production and cell viability (*Min et al., 2014*; *Mikulski and Santos-Aberturas, 2021*). Despite the existing and emerging contexts outlined above, knowledge about how rigid walls might modify the kinetics, kinematics, fluid flow and mixing, and theoretical description of a strongly confined microalga such as CR (or any other microswimmer) is scarce. All studies prior to ours have exclusively focused on the effect of boundaries on CR dynamics in PDMS chambers or thin fluid films of height $H \gtrsim 14\,\mu m$ (*Jeanneret et al., 2019*; *Ostapenko et al., 2018*; *Guasto et al., 2010*), that is, for weak confinement, $D/H < 1$.

Here, we present the first experimental measurements of the flagellar waveform, motility, and flow fields of *strongly confined* CR cells placed in between two *hard* glass walls ~10 μm apart ($D/H \gtrsim 1$, denoted 'H10 cells'), and infer from them the effect of confinement on kinetics, energy dissipation, and fluid mixing due to the cells. We also measure the corresponding quantities for weakly confined cells placed in glass chambers of height $H = 30\,\mu m$ ($D/H \sim 0.3$, denoted 'H30 cells') for comparison. We find that the cell speed decreases significantly and the trajectory tortuosity increases with increasing confinement as we go from H30 to H10 cells.

Surprisingly, the beat-cycle averaged experimental flow field of strongly confined cells has opposite flow vorticity to that expected from the screened version of bulk flow (*Drescher et al., 2010*; *Guasto et al., 2010*). This counterintuitive result comes about because the close proximity of the walls greatly suppresses the motility of the organism and, consequently, the thrust force of the flagella is balanced primarily by the non-hydrodynamic contact friction from the walls. The reason being that the flagellar thrust is largely unaffected by the walls, whereas the hydrodynamic drag on the slowly moving cell body is readily seen to be far smaller. Understandably, theoretical predictions from the source-dipole description of strongly confined swimmers do not account for this vortex flow inversion because they include only hydrodynamic stresses (*Brotto et al., 2013*; *Mathijssen et al., 2016*). We complement our experimental results with a simple theoretical description of the strongly confined microswimmer flows using a quasi-2D steady Brinkman approximation to the Stokes equation (*Brinkman, 1949*), instead of the complicated method of recursive images using Hankel transforms (*Liron and Mochon, 1976*; *Mathijssen et al., 2016*). Solving this equation, we demonstrate that the vortex flow inversion in strong confinement is well described as arising from a pair of like-signed force densities localized with a Gaussian spread around the approximate flagellar positions rather than the conventional three overall neutral point forces for CR (*Drescher et al., 2010*). We also show that under strong confinement there is enhanced fluid transport and mixing despite higher drag due to the walls.

## Results

### Experimental system

Synchronously grown wild-type CR cells (strain CC 1690) swim in a fluid medium using the characteristic breaststroke motion of two $\sim 11\,\mu m$ long anterior flagella with beat frequency $\nu_b \sim 50 - 60$ Hz.

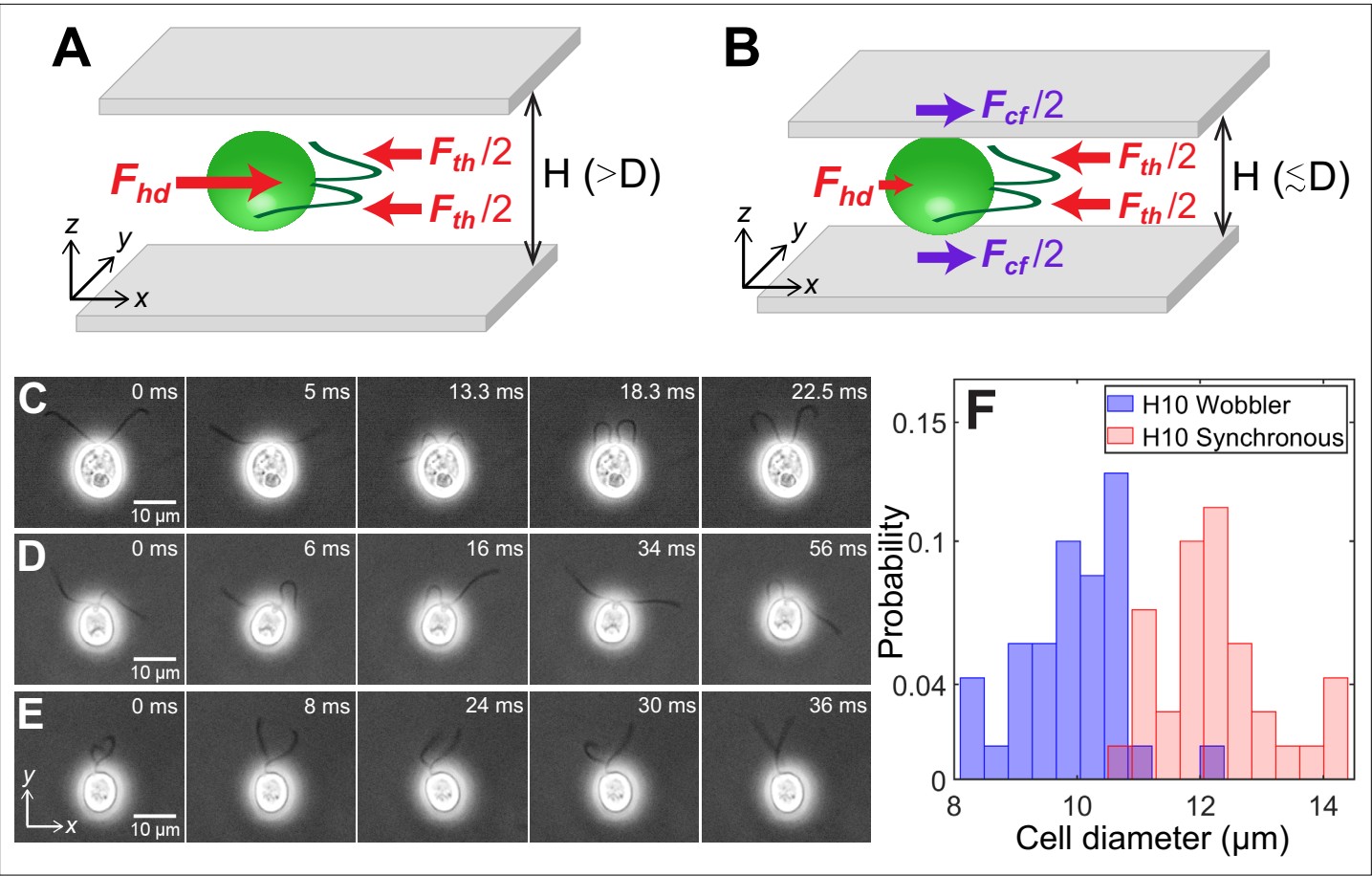

**Figure 1.** Cell size affects forces acting on confined microswimmers. Schematics of the forces exerted by a *Chlamydomonas* cell (green) swimming along the *x*-axis in between two glass plates separated by height, *H* under (**A**) weak confinement where the cell's body diameter, $D < H$, and (**B**) strong confinement where $D \gtrsim H$. Solid arrows represent local forces exerted by the cell on the surrounding medium. $\mathbf{F}_{th}$ and $F_{hd}$ are the propulsive thrust distributed equally between the two flagella and hydrodynamic drag due to the cell body, respectively. $F_{cf}$ is the contact friction with the strongly confining walls (**B**). Time lapse images of CR cells swimming in a quasi-2D chamber of height $H = 10\,\mu m$ with (**C**) synchronously beating flagella with $\nu_b \sim 39\,Hz$ ($D \sim 13.2\,\mu m$); (**D**) asynchronously beating flagella ($D \sim 9.9\,\mu m$); and (**E**) paddler type flagellar beat ($D \sim 9.7\,\mu m$). The cell bodies in (**D**) and (**E**) wobble due to their irregular flagellar beat pattern and are called 'Wobblers'. (**F**) Histogram of cell body diameter in the chamber of $H = 10\,\mu m$ (number of cells, $N = 70$). Synchronously beating cells ($N = 34$) typically have larger diameter than Wobblers ($N = 36$) and thus the H10 Synchronous cells with $D/H \gtrsim 1$ are strongly confined. Raw data is available in *Figure 1—source data 1*.

The online version of this article includes the following figure supplement(s) for figure 1:

**Source data 1.** Source data for *Figure 1F*.

These cells are introduced into rectangular quasi-2D chambers (area, 18 mm × 6 mm) made up of a glass slide and coverslip sandwich with double tape of thickness $H = 10/30\,\mu m$ as spacer. Passive 200 nm latex microspheres are added as tracers to the cell suspension for measuring the fluid flow using particle-tracking velocimetry. We use high-speed phase-contrast imaging at ~500 frames/s and ×40 magnification to capture flagellar waveform and cellular and tracer motion at a distance *H*/2 from the solid walls. The detailed experimental procedure is described in Materials and methods.

## Mechanical equilibrium of confined cells

The net force and torque on microswimmers, together with the ambient medium and boundaries, can be taken to be zero as gravitational effects are negligible in the case of CR for the range of length scales considered (*Drescher et al., 2010*; *Brennen and Winet, 1977*; *Pedley and Kessler, 1992*; *Elgeti et al., 2015*; *Mathijssen et al., 2016*). The two local forces exerted by any dipolar microswimmer on the surrounding fluid are flagellar propulsive thrust $\mathbf{F}_{th}$ and cell body drag $\mathbf{F}_{hd}$. They balance each other completely for any swimmer in an unbounded medium (*Lauga and Powers, 2009*; *Goldstein, 2015*)

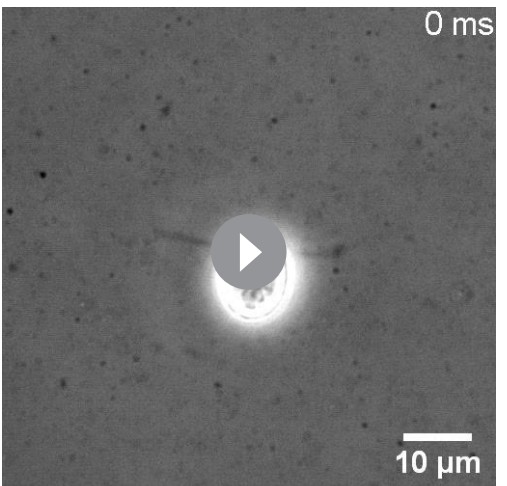

**Video 1.** Video of a strongly confined *Chlamydomonas* cell swimming with synchronous beat in the presence of tracers. High-speed video microscopy of a strongly confined swimmer (synchronously beating *Chlamydomonas* cell in $H = 10$ µm chamber) in the presence of tracer particles at 500 frames/s. This phase-contrast video clearly shows the synchronous breaststroke and planar beating of flagella with intermittent phase slips. This is the representative cell whose flow field is shown in Figure 3C. The direction of vortex flow is evident from the tracers' motion.
https://elifesciences.org/articles/67663/figures#video1

**Video 2.** Video of wobbling *Chlamydomonas* cells with asynchronous or paddling flagellar beat. Flagellar waveform of *Chlamydomonas* cells in $H = $ 10 µm chamber with wobbling cell body, that is, H10 Wobblers. The video is divided into three parts. The first part shows the asynchronous and planar flagellar beat of a cell which leads to a wobbling motion of the cell body. The second part shows the distinctive paddling flagellar beat of a cell, anterior to the cell body. Here, the flagellar beat plane is perpendicular to the imaging *x–y* plane and one of the flagella is mostly out of focus. In both these cases, the cell bodies wobble due to their irregular flagellar beat pattern. The third part shows a representative H10 Wobbler which switches from paddling beat to an asynchronous one.
https://elifesciences.org/articles/67663/figures#video2

and approximately in weak confinement between two hard walls (*Figure 1A*). In these regimes, CR is the classic example of an active puller where the direction of force dipole due to thrust and drag are such that the cell draws in fluid along the propulsion axis (*x*-axis in *Figure 1A*) and ejects it in the perpendicular plane (*Lauga and Powers, 2009*). CR is described well by three point forces or Stokeslets (*Drescher et al., 2010*) as in *Figure 1A* because the thrust is spatially extended and distributed equally between the two flagella. However, microswimmers in strong confinement between two closely spaced hard walls, $D/H \gtrsim 1$, are in a regime altogether different from bulk because the close proximity of the cells to the glass walls results in an additional drag force $\mathbf{F}_{cf}$ (*Figure 1B*). Therefore, the flagellar thrust is balanced by the combined drag due to the cell body and the strongly confining walls (*Figure 1B*).

## Size polydispersity, confinement heterogeneity, and consequences for flagellar waveform and motility

We define the degree of confinement of the CR cells as the ratio $D/H$ of cell body diameter to chamber height. CR cells in chambers of height $H = 30 \, \mu m$ are always in weak confinement as the cell diameter varies within $D \sim 8 - 14 \, \mu m < H$. However, this dispersity in cell size becomes significant when CR cells are swimming within quasi-2D chambers of height, $H = 10 \, \mu m$. Here, the diameter of individual cell is crucial in determining the character – weak or strong – of the confinement and, as a consequence, the forces acting on the cell. Below, we illustrate how the cell size determines the type of confinement in this regime through measurements of flagellar waveform and cell motility.

CR cells confined to swim in $H = 10 \, \mu m$ chambers show three kinds of flagellar waveform: (1) synchronous breaststroke and planar beating of flagella interrupted by intermittent phase slips ('H10 Synchronous', *Figure 1C*, *Video 1*); (2) asynchronous and planar flagellar beat over large time periods (*Figure 1D*, *Video 2*); and (3) a distinctive paddling flagellar beat wherein flagella often wind around each other and paddle irregularly anterior to the cell with their beat plane oriented away from the *x–y*

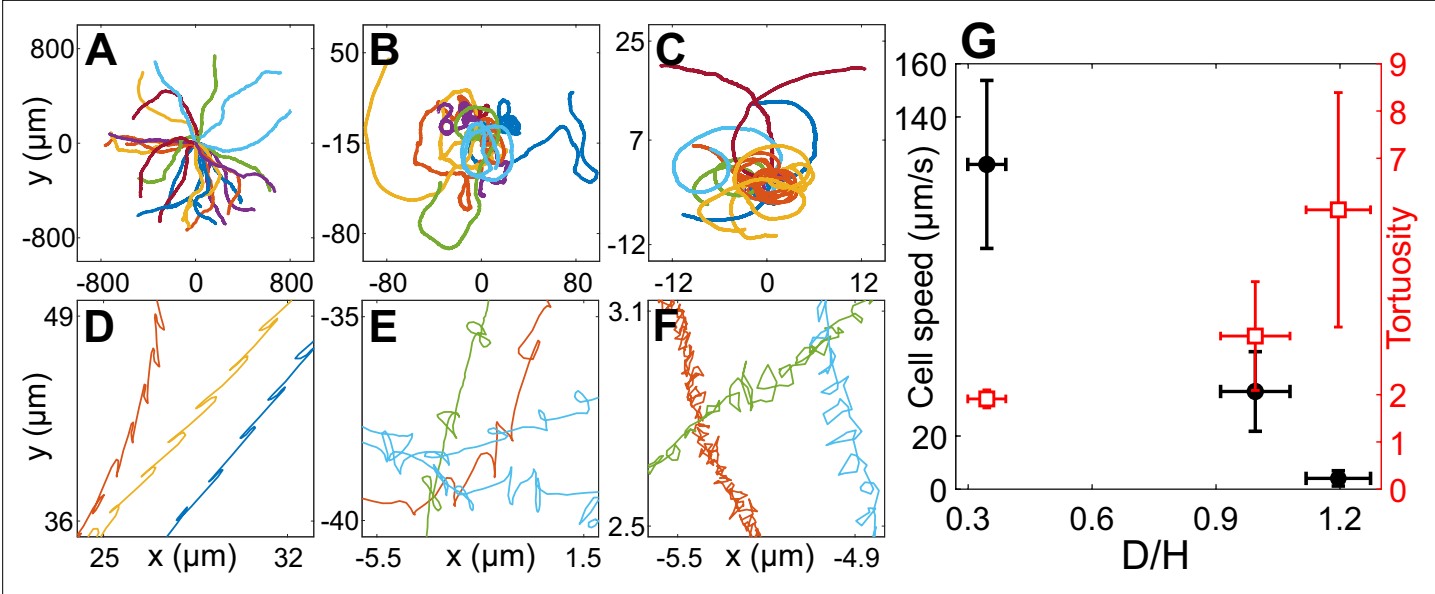

**Figure 2.** Cell motility in confinement. Representative trajectories of *Chlamydomonas reinhardtii* (CR) cells in (**A**) $H = 30\,\mu m$ ($N = 25$), (**B**) $H = 10\,\mu m$, Wobblers ($N = 13$); (**C**) $H = 10\,\mu m$, Synchronous cells ($N = 17$). All of these trajectories lasted for 8.2 s and their initial positions are shifted to origin. (**D**), (**E**) and (**F**) are the zoomed in trajectories of (**A**), (**B**) and (**C**), respectively. (**G**) Cell speed (circles) and tortuosity of trajectories (squares) as a function of the degree of confinement, $D/H$ ($N = 52, 35, 23$ for H30, H10 Wobbler and H10 Synchronous, respectively). The error bars in the plot correspond to standard deviation in diameter (*x*-axis), cell speed and tortuosity (*y*-axes) due to the heterogeneous population of cells. Raw data are available in *Figure 2—source data 1–4*.

The online version of this article includes the following source data for figure 2:

**Source data 1.** Source data for *Figure 2A*.

**Source data 2.** Source data for *Figure 2B*.

**Source data 3.** Source data for *Figure 2C*.

**Source data 4.** Source data for *Figure 2G*.

plane (*Figure 1E*, *Video 2*). While both synchronous and asynchronous beats are typically observed for CR in bulk (*Polin et al., 2009*) and weak confinement of $30\,\mu m$, the paddler beat is associated with calcium-mediated mechanosensitive shock response of the flagella to the chamber walls (*Fujiu et al., 2011*). The cell body wobbles for both asynchronous and paddler beat of cells (*Figure 1D, E*) and often the flagellar waveform in a single CR switches between these two kinds (*Video 2*). Hence, we collectively call them 'H10 Wobblers' (*Qin et al., 2015*).

We correlate the Synchronous and Wobbler nature of cells to their body diameter (*Figure 1F*). The mean projected diameter in the image plane of Synchronous cells ($D = 12.28 \pm 0.94\,\mu m$, number of cells, $N = 34$) is larger than that of Wobblers ($D = 9.92 \pm 0.85\,\mu m$, $N = 36$). Hence, the former's cell body is squished and *strongly confined* in $H = 10\,\mu m$ chamber in comparison with that of the latter. This leads to planar swimming of Synchronous cells, whereas Wobblers tend to spin about their body axis and trace out a near-helical trajectory which is a remnant of its behaviour in the bulk. Thus, the Wobblers likely compromise their flagellar beat into asynchrony and/or paddling over long periods, as a shock response, due to frequent mechanical interactions with the solid boundaries while rolling and yawing their cell body (*Fujiu et al., 2011*; *Choudhary et al., 2019*).

The motility of CR cells in $H = 30\,\mu m$ is similar to that in bulk and has the signature of back-and-forth cellular motion due to the recovery and power strokes of the flagella (*Figure 2A, D*). As confinement increases, the drag on the cells due to the solid walls increases and they trace out smaller distances with increasing twists and turns in the trajectory (*Figure 2A–F*). These phenomena can be quantitatively characterized by cell speed and trajectory tortuosity (Materials and methods) as a function of the degree of confinement of the cells (*Figure 2G*). Cellular speed decreases and tortuosity of trajectories increases with increasing confinement as we go from H30 → H10 Wobblers → H10 Synchronous cells. Notably, the cell speed $u$ decreases by 96% from H30 ($\langle u^{30} \rangle = 122.14 \pm 31.59\,\mu m/s$, $N = 52$) to H10

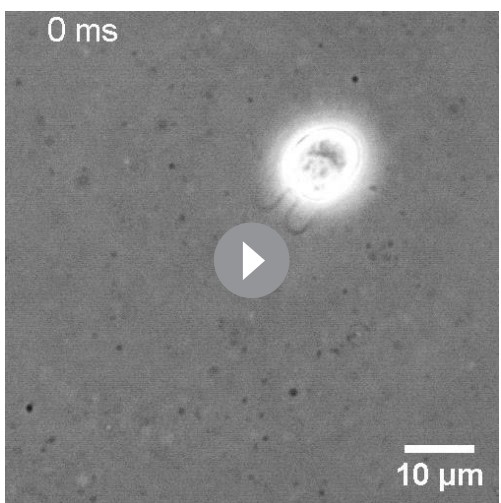

0 ms

10 μm

**Video 3.** Video of a weakly confined *Chlamydomonas* cell swimming in the presence of tracers. High-speed video microscopy of a weakly confined *Chlamydomonas* cell swimming in $H = 30$ μm chamber in the presence of tracer particles at 500 frames/s. This video shows the natural motility of cells in bulk where they spin about their body axis. The video starts with the cell and its flagella beating in the image plane. At ~90–180 ms, the flagellar beat of the cell is out of the image plane, when the cell body is rotating about its axis. The flow field is calculated only when the flagellar beat of the H30 cell is in the image plane, that is, for 0–90 and 180–252 ms for this particular video.
https://elifesciences.org/articles/67663/figures#video3

Synchronous swimmers ($\langle u^{10} \rangle = 4.07 \pm 2.88$ μm/s, $N = 23$). Henceforth, we equivalently refer to the H10 Synchronous CR as '*strongly confined*' or '*H10*' cells ($D/H \gtrsim 1$) and the H30 cells as '*weakly confined*' ($D/H < 1$).

We also note that the flagellar beat frequency of the strongly confined cells, $\nu_b^{10} \approx 51.58 \pm 7.62$ Hz (averaged over 210 beat cycles for $N = 20$) is similar to that of the weakly confined ones, $\nu_b^{30} \approx 55.27 \pm 8.22$ Hz (averaged over 194 beat cycles for $N = 20$). This is because even in the 10 μm chamber where the CR cell body is strongly confined, the flagella are beating far from the walls ($\sim 5$ μm) and almost unaffected by the confinement.

## Experimental flow fields

We measure the beat-averaged flow fields of H30 and H10 CR cells to systematically understand the effect of strong confinement on the swimmer's flow field. We determine the flow field for H30 cells only when their flagellar beat is in the *x–y* plane (*Video 3*) for appropriate comparison with planar H10 swimmers. *Figure 3A* shows the velocity field for H30 cells obtained by averaging ~178 beat cycles from 32 cells. It shows standard features of an unbounded CR's flow field (*Drescher et al., 2010*; *Guasto et al., 2010*), namely far-field four-lobe flow of a puller, two lateral vortices at 8–9 μm from cell's major axis, and anterior flow along the swimming direction till a stagnation point, 21 μm from the cell centre (*Figure 3B*). These near-field flow characteristics are quite well explained theoretically by a three-bead model (*Jibuti et al., 2017*; *Friedrich and Jülicher, 2012*; *Bennett and Golestanian, 2013*) or a three-Stokeslet model (*Drescher et al., 2010*), where the thrust is distributed at approximate flagellar positions between two Stokeslets of strength ($-1/2, -1/2$) balanced by a +1 Stokeslet due to viscous drag on the cell body (*Figure 1A*).

The flow field of a representative H10 swimmer ($u = 5.67 \pm 1.57$ μm/s, $\nu_b \sim 42.67 \pm 2.24$ Hz) is shown in *Figure 3C*, averaged over ~328 beat cycles. Strikingly, the vortices contributing dominantly to the flow in this strongly confined geometry are opposite in sign to those in the bulk (*Drescher et al., 2010*) or weakly confined case (H30, *Figure 3A*). This two-lobed flow is distinct from expectations based on the screened version of the bulk or three-Stokeslet flow, which is four-lobed (*Figure 3—figure supplement 1A*). Importantly, the far-field flow resembles a 2D source dipole pointing opposite to the swimmer's motion, which is entirely different from that produced by the standard source dipole theory of strongly confined swimmers (*Figure 3—figure supplement 1B*; *Brotto et al., 2013*; *Mathijssen et al., 2016*; *Jeanneret et al., 2019*). This is because the source-dipole treatment does not consider the possibility that the cells are squeezed by the walls, or in other words, it does not account for contact friction (*Brotto et al., 2013*; *Mathijssen et al., 2016*). Other significant differences from the bulk flow include front-back flow asymmetry, opposite flow direction posterior to the cell, distant lateral vortices (20 μm) and closer stagnation point (11 μm) (*Figure 3D*). All other H10 Synchronous swimmers, including the slowest ($u \sim 0.15$ μm/s) and the fastest ($u \sim 14$ μm/s) cells, show similar flow features. Even though the flow fields of H30 and H10 cells look strikingly different, the viscous power dissipated through the flow fields is nearly the same (Appendix 1.1).

A close examination suggests that the vortex contents of the flow fields of *Figure 3A* (H30) and *Figure 3C* (H10) are mutually compatible. The large vortices flanking the rapidly moving CR in H30 are shrunken and localized close to the cell body in H10 due to the greatly reduced swimming speed.

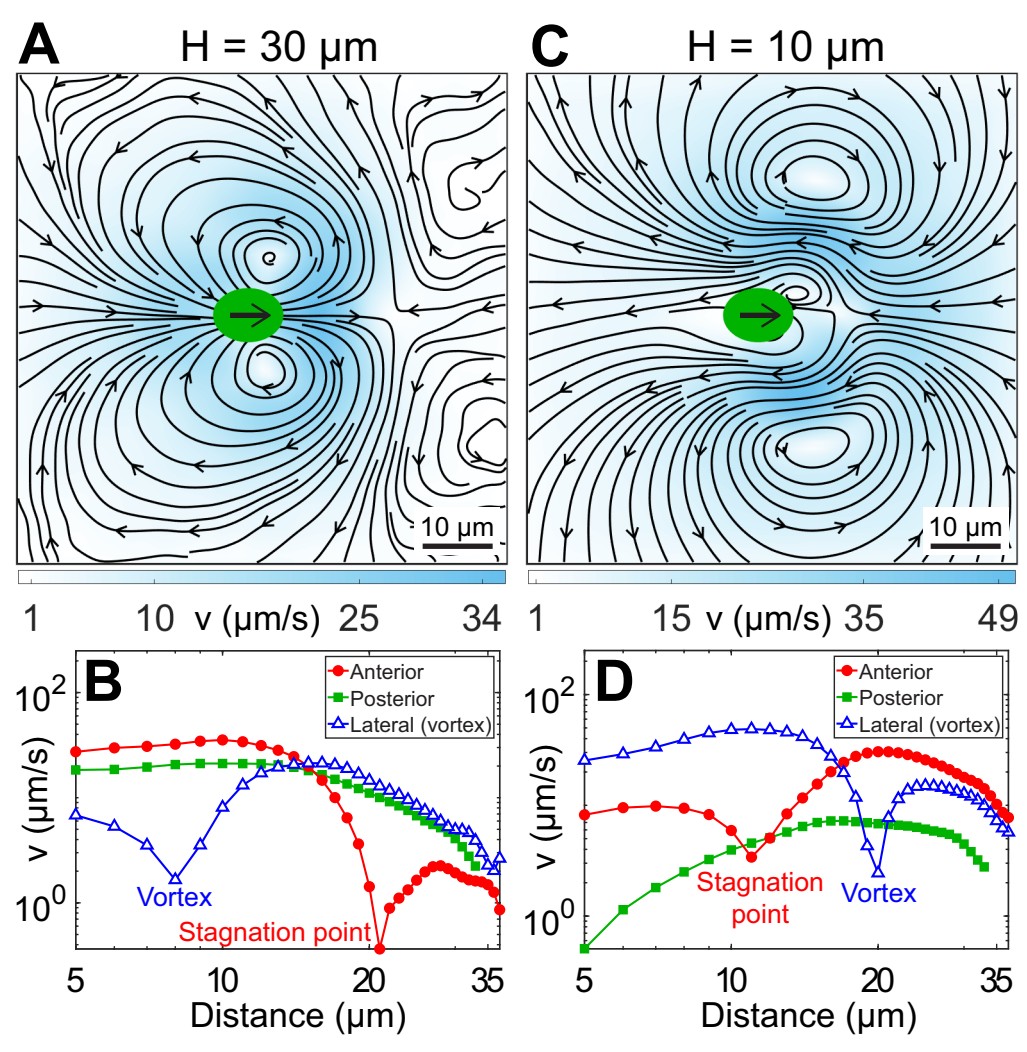

**Figure 3.** Experimental flow fields of *Chlamydomonas reinhardtii* (CR) cells in weak and strong confinement. Experimentally measured, beat-averaged flow fields in the *x*–*y* plane of synchronously beating CR cells swimming in (**A**) $H = 30\,\mu m$, and (**C**) $H = 10\,\mu m$. Black arrows on the cell body indicate that the cells are swimming to the right. Solid black lines indicate the streamlines of the flow in lab frame. The colourbars represent flow magnitude, *v*. (**B** and **D**) denote the speed variation in (**A** and **C**), respectively, along anterior, posterior, and lateral to the cell (where the vortices are present). Distances along anterior and posterior are measured along horizontal lines from the cell centre (0, 0); whereas the lateral (vortex) distances are measured along the vertical line passing through (*x*, *y*) = (2, 0) for (**B**) and (8, 0) for (**D**), respectively. Raw data are available in *Figure 3—source data 1* and *Figure 3— source data 2*.

The online version of this article includes the following source data and figure supplement(s) for figure 3:

**Source data 1.** Source data for *Figure 3A*.

**Source data 2.** Source data for *Figure 3C*.

**Figure supplement 1.** Expected flow fields of a strongly confined *Chlamydomonas reinhardtii* (CR) using conventional theoretical approaches.

The frontal vortices generated by flagellar motion now fill most of the flow field in H10. Generated largely during the power stroke of flagella, they are opposite in sense to the vortices produced by the moving cell body.

## Force balance on confined cells

In an unbounded fluid, the thrust $F_{th}$ exerted by the flagellar motion of the cell balances the hydrodynamic drag $F_{hd}$ on the moving cell body (*Figure 1A*). We assume this balance holds for the case of weak confinement (H30) as well. We estimate $|F_{hd}| = 3\pi\eta D u$ as the Stokes drag on a spherical cell body of diameter $D \simeq 10\,\mu m$ moving at speed $u$ through a fluid of viscosity $\eta = 1\,\text{mPa s}$ (*Goldstein, 2015*) which in the regime of weak confinement (H30), for a cell speed $u^{30} \approx 120\,\mu\text{m/s}$, is $F_{hd}^{30} \approx 11.31\,\text{pN}\,\hat{x}$, so that the corresponding thrust force is $F_{th}^{30} \approx -11.31\,\text{pN}\,\hat{x}$.

Given that CR operates at nearly constant thrust since $u \propto \eta^{-1}$ (*Qin et al., 2015*; *Rafaï et al., 2010*) and that the flagella of the H10 cell are beating far from the walls ($\sim 5\,\mu m$) with beat frequency and waveform similar to that of the H30 cell (*Videos 1 and 3*), we take the flagellar thrust force in strong confinement to be $F_{th}^{10} \approx F_{th}^{30} \approx -11.31\,\text{pN}\,\hat{x}$ as in weak confinement. This thrust is balanced by the total drag on the cell body. The cell speed, $u^{10} \approx 4\,\mu\text{m/s}$, is down by a factor of 30, and so is the hydrodynamic contribution to the drag if we assume the flow is the same as for the H30 geometry. Even if we take into account the tight confinement, and thus assume that the major hydrodynamic drag comes (*Brotto et al., 2013*; *Persat et al., 2015*; *Bhattacharya et al., 2005*) from a lubricating film of thickness $\delta = (H - D)/2 \ll D$ between the cell and each wall, the enhancement of drag due to the fluid, logarithmic in $\delta/D$ (*Bhattacharya et al., 2005*; *Ganatos et al., 2006*), cannot balance thrust for any plausible value of $\delta$.

The above imbalance drives the vortex flow inversion observed in *Figure 3C*, as will be shown later theoretically, and implies that the drag is dominated by the direct frictional contact between the cell body and the strongly confining walls, which we denote by $F_{cf}$. Force balance on the fluid element and rigid walls enclosing the CR in strong confinement requires $F_{th}^{10} + F_{hd}^{10} + F_{cf}^{10} = 0$ (*Figure 1B*). We know that the hydrodynamic drag under strong confinement is greater than $0.38\,\text{pN}$ (Stokes drag at $u^{10} \approx 4\,\mu\text{m/s}$), but lack a more accurate estimate as we do not know the thickness $\delta$ of the lubricating film. We can therefore say that the contact force $F_{cf}^{10} \lesssim 10.93\,\text{pN}\,\hat{x}$. Thus, the flagellar thrust works mainly against the non-hydrodynamic contact friction from the walls as expected due to the extremely low speed of the strongly confined swimmer.

## Theoretical model of strongly confined flow

We begin by using the well-established far-field solution of a parallel Stokeslet between two plates by Liron and Mochon in an attempt to explain the strongly confined CR's flow field (*Liron and Mochon, 1976*). However, the theoretical flow of Liron and Mochon decays much more rapidly than the experimental one and does not capture the vortex positions and flow variation in the experiment (Appendix 1.2 and *Appendix 1—figure 1*). This is because the Liron and Mochon approximation to the confined Stokeslet flow is itself singular and also the far-field limit of the full analytical solution, so it cannot be expected to accurately explain the near-field characteristics of the experimental flow (*Liron and Mochon, 1976*).

We therefore start afresh from the incompressible 3D Stokes equation, $-\boldsymbol{\nabla} p(\boldsymbol{r}) + \eta\nabla^2\boldsymbol{v}(\boldsymbol{r}) = 0$, $\nabla \cdot \boldsymbol{v}(\boldsymbol{r}) = 0$, where $p$ and $\boldsymbol{v}$ are the fluid pressure and velocity fields, respectively. Next, we formulate an effective 2D Stokes equation and find its point force solution. In a quasi-2D chamber of height $H$, we consider an effective description of a CR swimming in the $z = 0$ plane of the coordinate system with the first Fourier mode for the velocity profile along $z$, satisfying the no-slip boundary condition on the solid walls, $\boldsymbol{v}(x, y, z = \pm H/2) = 0$ (*Figure 4—figure supplement 1*). Therefore, the flow velocity varies as $\boldsymbol{v}(x, y, z) = \boldsymbol{v}^0(x, y)\cos(\pi z/H)$ (*Figure 4—figure supplement 1*), where $\boldsymbol{v}^0 = (v_x, v_y)$ is the flow profile in the swimmer's $x - y$ plane that is experimentally measured in *Figure 3* (*Fortune et al., 2021*). Substituting this form of velocity field in the Stokes equation we obtain its quasi-2D Brinkman approximation (*Brinkman, 1949*), which for a point force of strength $\boldsymbol{F}$ at the $z = 0$ plane, is

$$-\boldsymbol{\nabla}_{xy} p(\boldsymbol{r}) + \eta\left(\nabla_{xy}^2 - \frac{\pi^2}{H^2}\right)\boldsymbol{v}(\boldsymbol{r}) + \boldsymbol{F}\delta(\boldsymbol{r}) = 0, \quad \nabla_{xy} \cdot \boldsymbol{v}(\boldsymbol{r}) = 0 \tag{1}$$

where $p$ and $\boldsymbol{v} \equiv \boldsymbol{v}^0$ are the pressure and fluid velocity in the $x-y$ plane and $\nabla_{xy} = \partial_x \widehat{\boldsymbol{x}} + \partial_y \widehat{\boldsymbol{y}}$. We Fourier transform the above equation in 2D and invoke the orthogonal projection operator $\boldsymbol{O_k} = 1 - \widetilde{\boldsymbol{kk}}$ to annihilate the pressure term and obtain the quasi-2D Brinkman equation in Fourier space.

$$\boldsymbol{v_k} = \frac{\boldsymbol{O_k \cdot F}}{\eta \left( k^2 + \dfrac{\pi^2}{H^2} \right)} \tag{2}$$

We perform inverse Fourier transform on *Equation 2* in 2D for a Stokeslet oriented along the $x$-direction, $\boldsymbol{F} = F\widehat{\boldsymbol{x}}$ to obtain its flow field $\boldsymbol{v(r)}$ at the $z = 0$ plane (Appendix 1.3). This solution is identical to the analytical closed-form expression of *Pushkin and Bees, 2016*. We have already shown that superposing our Brinkman solution for the conventional three point forces at cell centre and flagellar positions of CR, which leads to the effective three-Stokeslet model in 2D, is an inappropriate description of the strongly confined flow (*Figure 3—figure supplement 1A*). This is not surprising at this point because the force imbalance between the flagellar thrust and hydrodynamic cell drag suggests that the cell is nearly stationary compared to the motion of its flagella. We utilize this experimental insight by superposing only two Stokeslets of strength $-1/2\,\widehat{\boldsymbol{x}}$ each at approximate flagellar positions $(x_f, \pm y_f) = (6, \pm 11)\,\mu m$ to find qualitatively similar streamlines and vortex flows (*Figure 4A*) as that of the experimental flow field (*Figure 3C*). However, this theoretical 'two-Stokeslet Brinkman flow' (*Figure 4A*) decays faster than the experiment as shown in the quantitative comparison of these two flows in *Figure 4B* and *Figure 4—figure supplement 2A,B*. The root mean square deviation (RMSD) between these two flows in $v_x$, $v_y$, and $|v|$ are 20.3%, 14.2%, and 22.6%, respectively (see Materials and methods for RMSD definition).

With the experimental streamlines and vortices well described by a two-Stokeslet Brinkman model, we now explain the slower flow variation in experiment. Strongly confined experimentally observed flow is mostly ascribed to the flagellar thrust, as described above. Clearly, a delta-function point force will not be adequate to describe the thrust generated by flagellar beating as they are slender rods of length $L \sim 11\,\mu m$ with high aspect ratio. We, therefore, associate a 2D Gaussian source $g(\boldsymbol{r}) = \dfrac{e^{-r^2/2\sigma^2}}{2\pi\sigma^2}$ of standard deviation $\sigma$, to *Equation 1* instead of the point source $\delta(\boldsymbol{r})$, in a manner similar to the regularized Stokeslet approach (*Cortez et al., 2005*). Thus, the quasi-2D Brinkman equation in Fourier space (*Equation 2*) for a Gaussian force $\boldsymbol{F}g(\boldsymbol{r})$ becomes,

$$\boldsymbol{v_k} = \frac{\boldsymbol{O_k \cdot F}}{\eta \left( k^2 + \dfrac{\pi^2}{H^2} \right)} e^{-k^2\sigma^2/2}. \tag{3}$$

Superposing the inverse Fourier transform of the above equation for two sources of $\boldsymbol{F} = (-1/2,\ -1/2)\widehat{\boldsymbol{x}}$ at $(x_f, \pm y_f) = (6, \pm 11)\,\mu m$ with $\sigma \sim L/2 = 5\mu m$, we obtain the theoretical flow shown in *Figure 4C*. RMSD in $v_x$, $v_y$, and $|v|$ between this theoretical flow and those of the experimental one (*Figure 3C*) are 7.8%, 9%, and 8.3%, respectively. Comparing these two flows along representative radial distances from the cell centre as a function of polar angle show a good agreement (*Figure 4D* and *Figure 4—figure supplement 2C, D*). Notably, *Figure 4C*, that is, the 'two-Gaussian Brinkman flow', has captured the flow variation and most of the experimental flow features accurately. Specifically, these are the lateral vortices at 20 μm and an anterior stagnation point at 13 μm from cell centre. The only limitation of this theoretical model is that it cannot account for the front-back asymmetry of the strongly confined flow, as is evident from *Figure 4D* for the polar angles 0 or 2π and π which correspond to front and back of the cell, respectively. This deviation is more pronounced in the frontal region as the cell body squashed between the two solid walls mostly blocks the forward flow from reaching the cell posterior. Thus, the no-slip boundary on the cell body needs to be invoked to mimic the front-back flow asymmetry, which is a more involved analysis due to the presence of multiple boundaries and can be addressed in a follow-up study.

Now that we have explained the flow field of CR in strong confinement, we test our quasi-2D Brinkman theory in weak confinement, $H = 30\,\mu m$, where the thrust and drag forces almost balance each other. Hence, we use the conventional three-Stokeslet model for CR, but with a Gaussian

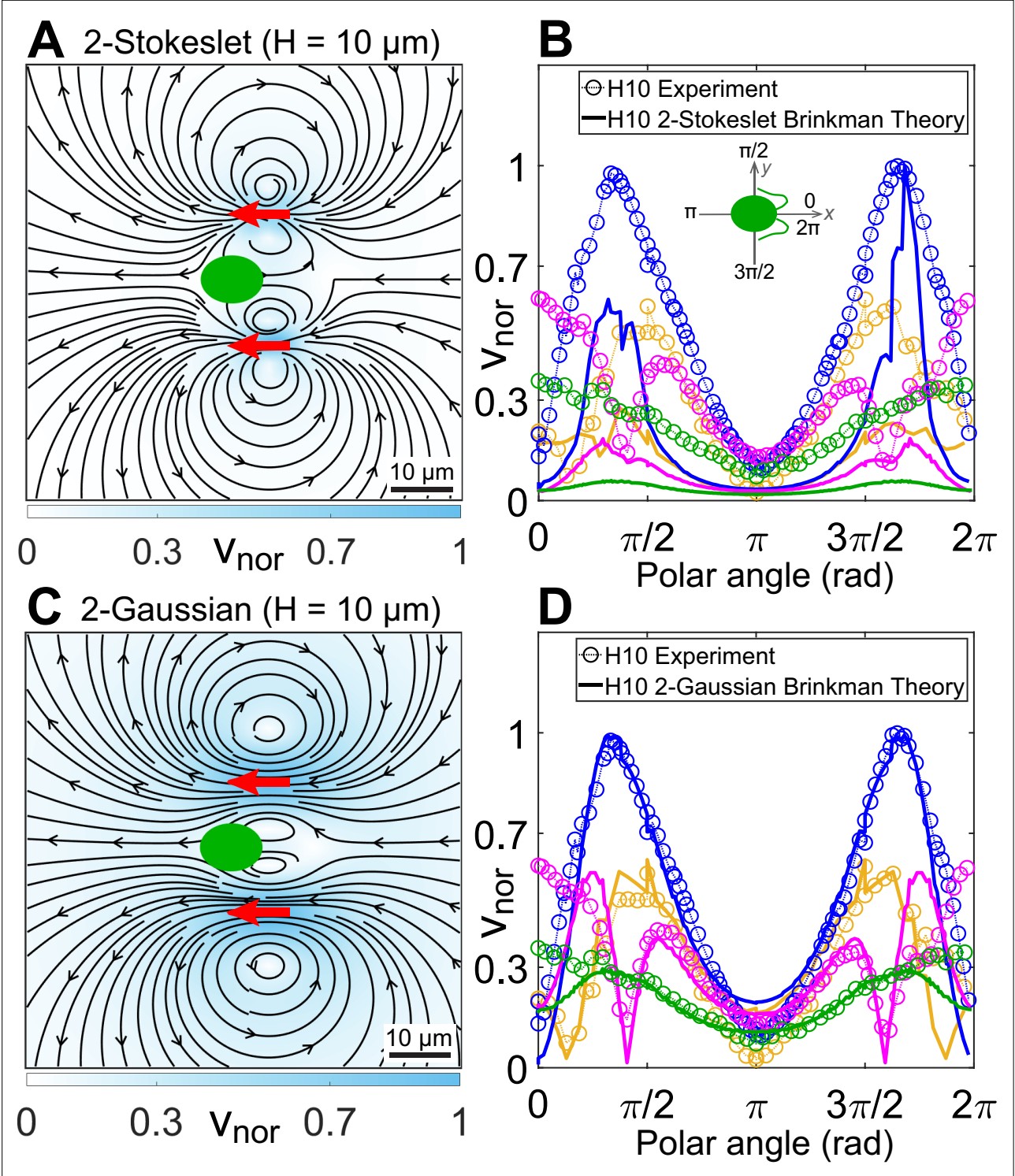

**Figure 4.** Theoretical flow fields in strong confinement. Theoretically computed flow fields for (**A**) two Stokeslets and (**C**) two Gaussian forces, both positioned at $(6, \pm11)\,\mu m$ (red arrows) using the quasi-2D Brinkman equation for $H = 10\,\mu m$ at the $z = 0$ plane. The colourbars represent flow magnitudes normalized by their maximum, $v_{nor}$. (**B, D**) Comparison of normalized experimental flow of the *Chlamydomonas reinhardtii* (CR) in $H = 10\,\mu m$ (*Figure 3C*) with theoretical flow fields (**A**) and (**C**), respectively, along representative radial distances, $r$, from the cell centre as a function of polar angle. Inset of (**B**) shows the convention used for polar angle. Plots for each $r$ denote the flow magnitudes for those grid points which lie in the radial gap $(r, r + 1)\,\mu m$; $r\,(\mu m)$ = 7 (yellow), 13 (blue), 20 (magenta), and 30 (green). Raw data are available in *Figure 4—source data 1* and *Figure 4—source data 2*.

*Figure 4 continued on next page*

*Figure 4 continued*

The online version of this article includes the following source data and figure supplement(s) for figure 4:

**Source data 1.** Source data for *Figure 4A*.

**Source data 2.** Source data for *Figure 4C*.

**Figure supplement 1.** Schematic of velocity profile along the confining direction.

**Figure supplement 2.** Comparison in the direction of flow fields between experiment and theory.

**Figure supplement 3.** Theoretical flow field in weak confinement.

distribution for each point force. We, therefore, superpose the solution of *Equation 3* for three-Gaussian forces representing the cell body and two flagella in $H = 30\,\mu m$. The resulting flow field (*Figure 4—figure supplement 3*) matches qualitatively with the experimental flow field of CR in weak confinement (*Figure 3A*). This deviation is expected in weak confinement, $D/H \sim 0.3$, because the quasi-2D theoretical approximation is mostly valid at $D/H \gtrsim 1$, even though RMSD in $v_x$, $v_y$, and $|v|$ remain in the low range at 11.4%, 11.2%, and 13.8%, respectively.

Together, the experimental and theoretical flow fields show that the contact friction from the walls reduces the force-dipolar swimmer in bulk or weak confinement (H30) to a force-monopole one in strong confinement (H10).

## Enhancement of fluid mixing in strong confinement

The photosynthetic alga CR feeds on dissolved inorganic ions/molecules such as phosphate, nitrogen, ammonium, and carbon dioxide from the surrounding fluid in addition to using sunlight as the major source of energy (*Tam and Hosoi, 2011*; *Kiørboe, 2008*). Importantly, nitrogen and carbon are limiting macronutrients to algal growth and metabolism (*Khan et al., 2018*; *Short et al., 2006*; *Kiørboe, 2008*). For example, dissolved carbon dioxide in the surrounding fluid contains the carbon source essential for photosynthesis and acts as pH buffer for optimum algal growth. It is widely known that flagella-generated flow fields help in uniform distribution of these dissolved solute molecules through fluid mixing and transport which have a positive influence on the nutrient uptake of osmotrophs like CR (*Kiørboe, 2008*; *Tam and Hosoi, 2011*; *Ding et al., 2014*; *Short et al., 2006*; *Leptos et al., 2009*; *Kurtuldu et al., 2011*). This is even more important for the strongly confined CR cells as they cannot move far enough to outrun diffusion of nutrient molecules because of slow swimming speed.

We first calculate the flow-field-based Péclet number, $Pe = Vl_V/D_S$ where $V$ and $l_V$ are the flow-speed and diameter of the flagellar vortex, and $D_S$ is the solute diffusivity in water, as the standard measure to characterize the relative significance of advective to diffusive transport. Using the experimentally measured flow data from *Figure 3* and $D_S \approx 10^{-9}$ m$^2$/s (*Shapiro et al., 2014*; *Kiørboe, 2008*; *Tam and Hosoi, 2011*), we compute the Péclet numbers for the weakly and strongly confined cell to be $Pe^{30} \approx 0.5$ and $Pe^{10} \approx 2$, respectively (see *Appendix 1—table 1* and Appendix 1.4). These numbers suggest that flow-field-mediated advection does not completely dominate, but nevertheless can play a role in nutrient uptake for small biological molecules along with diffusion-mediated transport, especially for the strongly confined cell. However, it is evident from the recorded videos of weakly and strongly confined cell suspensions that the tracers are advected more in the H10 than in the H30 chamber (*Videos 1 and 3*). Hence, we attempt to quantify the observed differences in fluid mixing through correlation in flow velocity and displacement of passive tracers by the swimmers.

We calculate the normalized spatial velocity–velocity correlation function of the flow fields, $C_{vv}(R) = \dfrac{\langle v(r) \cdot v(r + R) \rangle}{\langle v(r) \cdot v(r) \rangle}$ to estimate the enhancement of fluid mixing in strong confinement (*Figure 5A*). The fluctuating flow field has a correlation length, $\lambda = 13.2\,\mu m$ for the strongly confined H10 flow, which is 37.5% higher than the weakly confined flow in $H = 30\,\mu m$ ($\lambda = 9.6\,\mu m$), even though the cell is swimming very slowly in strong confinement. This observation is complementary to the experiments of *Kurtuldu et al., 2011* where enhanced mixing is observed for active CR suspensions in 2D soap films compared to those in 3D unconfined fluid (*Leptos et al., 2009*). In their case, the reduced spatial dimension leads to long-ranged flow correlations due to the stress-free boundaries (the force-dipolar flow reduces from $v \sim r^{-2}$ in 3D to $v \sim r^{-1}$ in 2D). In our case, strong confinement reduces the force-dipolar swimmer in H30 to a force-monopole one in H10 (as shown in the previous

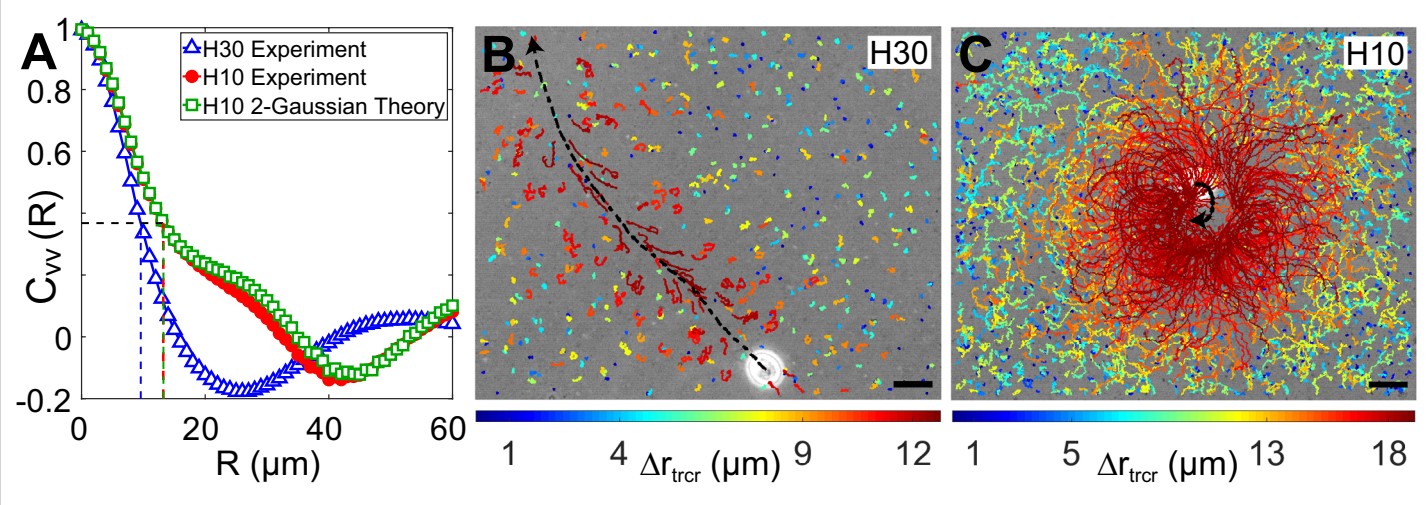

**Figure 5.** Correlation in fluid flow and tracer displacements. (**A**) Normalized radial velocity–velocity correlation function, $C_{vv}(R)$, of flow fields in *Figures 3A, C and 4C*. The dashed vertical lines denote the correlation length scales for the flows, $\lambda = 9.6\,\mu m$ (H30) and 13.2 μm (H10, both experiment and theory), where the correlation function decays to $1/e$ (horizontal dashed line). (**B** and **C**) Snapshots showing passive tracer trajectories (coloured) due to a *Chlamydomonas reinhardtii* (CR) cell (white) swimming along the black dashed arrow in $H = 30\,\mu m$ and $H = 10\,\mu m$, respectively. The H30 swimmer ($u = 121\,\mu m/s$) passes through the field of view within 1.3 s whereas the H10 cell ($u = 3\,\mu m/s$) traces a semicircular trajectory staying in the field of view for the recording time of 8.2 s. The tracer trajectories are colour coded, according to the colourbar below, based on their maximum displacement, $\Delta r_{trcr}$, during a fixed lag time of $\Delta t = 0.2\,s$ (~10 flagellar beat cycles). Scale bars, 15 μm. Raw data is available in *Figure 5—source data 1*.

The online version of this article includes the following source data and figure supplement(s) for figure 5:

**Source data 1.** Source data for *Figure 5A*.

**Figure supplement 1.** Mean-squared displacement (MSD) of tracers.

section). This leads to longer correlation length scales in the flow velocity, which implies an increased effective diffusivity (scaling $\sim V_{rms}\lambda$ for a velocity field with RMS value $V_{rms}$) of the fluid particles on time scales $\gg \lambda/V_{rms}$, in strong confinement.

Next, we measure the displacement of the passive tracer particles when a single swimmer passes through the field of view (179 μm × 143 μm) in our experiments. The H30 swimmers are fast and therefore pass through this field of view in ~1-1.4 s (*Figure 5B*), whereas the slow-moving H10 swimmers stay in the field of view for the maximum recording time of ~8 s (*Figure 5C*). As the swimmer moves within the chamber, it perturbs the tracer particles. The trajectories of these tracer particles involve both Brownian components and large jumps induced by the motion and flow field of these swimmers. We colour code the tracer trajectories based on their maximum displacement, $\Delta r_{trcr}$, during a fixed lag time of $\Delta t = 0.2\,s$ (~10 flagellar beat cycles) (*Figure 5B,C*). The tracer trajectories close to the swimming path of the representative H30 swimmer (black dashed arrow) are mostly advected by the flow whereas those far away from the cell involve mostly Brownian components (*Figure 5B*). However, a majority of the tracers in the full field of view are perturbed due to the H10 flow, those in the close vicinity being mostly affected (*Figure 5C*). Their advective displacements are larger than that of the tracers due to H30 flow (see the colourbar below).

We define the spatial range to which a swimmer motion advects the tracers — radius of influence, $R_{ad}$ — to be approximately equal to the lateral distance from the cell's swimming path (black dashed arrow) where the tracer displacements decrease to ~20% of their maxima (dark orange trajectories). The region of influence for the H30 cell is a cylinder of radius $R_{ad} \approx 15\,\mu m$ with the cell's swimming path as its axis (*Figure 5B*) and that for the H10 cell is a sphere of radius $R_{ad} \approx 35\,\mu m$ centred on the slow swimming cell's trajectory (*Figure 5C*). That is, the radius of influence of the H10 flow is higher than the H30 one, which corroborates the longer velocity correlation length scale in strong confinement. We also measure the mean-squared displacement (MSD) of the tracers to quantify the relative increment in the advective transport of the H10 flow with respect to the H30 one. We calculate the MSD of approximately 500 tracers in the whole field of view for each video where a single cell is passing through it and then ensemble average over six such videos (*Figure 5—figure supplement*

*1*). These plots with a scaling $\langle \Delta r_{trcr}^2 \rangle \propto \Delta t^\alpha$ show a higher MSD exponent in H10 ($\alpha \simeq 1.55$) than H30 ($\alpha \simeq 1.25$) indicating enhanced anomalous diffusion in strong confinement. Together, *Figure 5* shows that the fluid is advected more in strong confinement leading to enhanced fluid mixing and transport. In other words, the opposite vortical flows driven by flagellar beating in strong confinement help in advection-dominated dispersal of nutrients, air and $CO_2$ in the surrounding fluid, thereby aiding the organism to avail itself of more nutrients for growth and metabolism.

## Discussion

Our results show that a prototypical puller-type of microswimmer like CR, when squeezed between two solid walls with a gap that is narrower than its size, has a remarkably different motility and flow field from those of a bulk swimmer. In this regime of strong confinement, the cells experience a non-hydrodynamic contact friction that is large enough to decrease their swimming speed by 96%. Consequently, their effect on the fluid is dominantly through the flagella, which pull the fluid towards the organism and therefore, the major vortices in the associated flow field have vorticity opposite to that observed in bulk or weak confinement. This leads to an increased mixing and transport through the flow in strong confinement. These experimental results, which arise due to mechanical friction from the walls and not due to any behavioural change, establish that confinement not only alters the hydrodynamic stresses but also modifies the swimmer motility which in turn impacts the fluid flows. This coupling between confinement and motility is typically ignored in theoretical studies because the focus tends to be on the effect of confining geometry on flow fields induced by a given set of force generators (*Brotto et al., 2013*; *Mathijssen et al., 2016*), which is appropriate for weak confinement, whereas strong confinement alters the complexion of forces generating the flow. Recent experimental reports have not observed the effect we discuss because they confine CR in chambers of height greater than the cell size ($D/H \lesssim 0.7$) (*Jeanneret et al., 2019*) where the stresses are mostly hydrodynamic and therefore their theoretical model is force free and different from ours (Appendix 1.5).

Our theoretical approach of using two like-signed Brinkman Stokeslets localized with a Gaussian spread on the propelling appendages can also be easily utilized to analyze flows of a dilute collection of strongly confined swimmers (Appendix 1.6 and *Appendix 1—figure 2*). Notably, the force-monopolar flow field of the strongly confined CR is similar to that of tethered microorganisms like *Vorticella* within the slide-coverslip experimental setup (*Pepper et al., 2009*; *O'Malley, 2011*). Therefore, our effective 2D theoretical model involving Brinkman Stokeslet is applicable to these contexts as well. However, one needs to account for the differences in ciliary beating (two-ciliary flow for CR whereas multi-ciliated metachronal waves for *Vorticella*) for a comprehensive description of the flow field closer to the organism (*Pepper et al., 2009*; *Ryu et al., 2016*).

We note that even though CR is known to glide on liquid-infused solid substrates through flagella-mediated adhesive interactions (*Sasso et al., 2018*), it has recently been shown that the strength of flagellar adhesion is sensitive to and switchable by ambient light (*Kreis et al., 2017*). Consequently, it is likely that CR in its natural habitat of rocks and soils would also utilize swimming in addition to gliding. Our quantitative analysis shows that despite the higher frictional drag due to the strongly confining walls, there is enhanced fluid mixing due to the H10 flow field. That is, the inverse vortical flows driven by the flagellar propulsive thrust help in advection-mediated transport of nutrients to the strongly confined microswimmer. This suggests that swimming is more efficient than gliding for CR under strong confinement (especially in low-light conditions), even though CR speeds are of the same order in both these mechanisms [$u_{glide} \sim 1\,\mu m/s$ (*Sasso et al., 2018*) and $u_{swim} \sim 4\,\mu m/s$]. We note that apart from the time-averaged flows, the oscillations produced in the flow ($v^{osc}$) due to the periodic beating of the flagella can play a role in fluid transport and mixing for both the H30 ($\nu_b \sim 55$ Hz, order of magnitude estimate of $v^{osc} \sim L \times 2\pi\nu_b \sim 3450\,\mu m/s$) and H10 ($\nu_b \sim 52$ Hz, $v^{osc} \sim 3270\,\mu m/s$) cells (*Guasto et al., 2010*; *Klindt and Friedrich, 2015*).

Finally, our experimental and theoretical methodologies are completely general and can be applied to any strongly confined microswimmer, biological or synthetic from individual to collective scales. Specifically, our robust and efficient description using point or Gaussian forces in a quasi-2D Brinkman equation is simple enough to implement and analyze confined flows in a wide range of active systems. We expect our work to inspire further studies on biomechanics and fluid mixing due to hard-wall confinement of concentrated active suspensions (*Kurtuldu et al., 2011*; *Pushkin and*

*Yeomans, 2014*; *Jin et al., 2021*). These effects can be exploited in realizing autonomous motion through microchannel for biomedical applications and in microfluidic devices for efficient control, navigation and trapping of microbes and synthetic swimmers (*Park et al., 2017*; *Karimi et al., 2013*; *Temel and Yesilyurt, 2015*).

## Materials and methods

### Surface modification of microspheres and glass surfaces

CR cells are synchronously grown in 12:12 hr light:dark cycle in Tris-Acetate-Phosphate (TAP + P) medium. This culture medium contains divalent ions such as $Ca^{2+}$, $Mg^{2+}$, $SO_4^{2-}$ which decrease the screening length of the 200 nm negatively charged microspheres, thereby promoting inter-particle aggregation and sticking to glass surfaces and CR's flagella. Therefore, the sulfate latex microspheres (S37491, Thermo Scientific) are sterically stabilized by grafting long polymer chains of polyethylene glycol (mPEG-SVA-20k, NANOCS, USA) with the help of a positively charged poly-l-lysine backbone (P7890, 15–30 kDa, Sigma) (*Mondal et al., 2020*). In addition, the coverslip and slide surfaces are also cleaned and coated with polyacrylamide brush to prevent non-specific adhesion of microspheres and flagella to the glass surfaces, prior to sample injection (*Mondal et al., 2020*).

### Sample imaging

Cell suspension is collected in the logarithmic growth phase within the first 2–3 hr of light cycle and re-suspended in fresh TAP + P medium. After 30 min of equilibration, the cells are injected into the sample chamber. The sample chamber containing cells and tracers is mounted on an inverted microscope (Olympus IX83/IX73) and placed under red light illumination (>610 nm) to prevent adhesion of flagella (*Kreis et al., 2017*) and phototactic response of CR (*Sineshchekov et al., 2002*). We let the system acclimatize in this condition for 40 min before recording any data. All flow field data, flagellar waveform and cellular trajectory (except for *Figure 2A*) are captured using a ×40 phase objective (Olympus, 0.65 NA, Plan N, Ph2) coupled to a high-speed CMOS camera (Phantom Miro C110, Vision Research, pixel size = 5.6 μm) at 500 frames/s. As CR cells move faster in $H = 30\,\mu m$ chamber, a 8.2 s long trajectory cannot be captured at that magnification. So we used a ×10 objective in bright field (Olympus, 0.25 NA, PlanC N) connected to a high-speed camera of higher pixel length (pco.1200hs, pixel size = 12 μm) at 100 frames/s to capture 8.2 s long trajectories of H30 cells (*Figure 2A*).

Our observations are consistent across CR cultures grown on different days and cultures inoculated from different colonies of CR agar plates. We have prepared at least 15–18 samples of dilute CR suspensions from eight different days/batches of cultures, each for chambers of height 10 and 30 μm. Our imaging parameters remain same for all observations. We also use the same code, which is verified from standard particle-tracking videos, for tracking all the cells. We modify the cell tracking code to track the tracer motion for calculating the flow-field data.

### Height measurement of sample chamber

We use commercially available double tapes of thickness 10 and 30 μm (Nitto Denko Corporation) as spacer between the glass slide and coverslip. To measure the actual separation between these two surfaces, we stick 200 nm microspheres to a small strip (18 mm × 6 mm) on both the glass surfaces by heating a dilute solution of microspheres. Next, we inject immersion oil inside the sample chamber to prevent geometric distortion due to refractive index mismatch between objective immersion medium and sample. The chamber height is then measured by focusing the stuck microspheres on both surfaces through a ×60 oil-immersion phase objective (Olympus, 1.25 NA). We find the measured chamber height for the 10 μm spacer to be 10.88 ± 0.68 μm and for the 30 μm spacer to be 30.32 ± 0.87 μm, from eight different samples in each case.

### Particle-tracking velocimetry

The edge of a CR cell body appears as a dark line (*Figure 1C–E*) in phase-contrast microscopy and is detected using ridge detection in ImageJ (*Wagner and Hiner, 2017*). An ellipse is fitted to the pixelated CR's edge and the major axis vertex in between the two flagella is identified through custom-written MATLAB codes (refer to *Source code 1*). The cell body is masked and the tracers' displacement in between two frames (time gap, 2 ms) are calculated in the lab frame using standard

MATLAB tracking routines (*Blair and Dufresne, 2008*). The velocity vectors obtained from multiple beat cycles are translated and rotated to a common coordinate system where the cell's major axis vertex is pointing to the right (*Figure 3A, C*). Outliers with velocity magnitude more than six standard deviations from the mean are deleted. The resulting velocity vectors from all beat cycles (including those from different cells in $H = 30\,\mu m$) are then placed on a mesh grid of size 2.24 μm × 2.24 μm and the mean at each grid point is computed. The gridded velocity vectors are then smoothened using a 5 × 5 averaging filter. Furthermore, for comparison with theoretical flow, the x and y components of the velocity vectors are interpolated on a grid size of 1 × 1 μm². Streamlines are plotted using the '*streamslice*' function in MATLAB.

## Trajectory tortuosity

Tortuosity characterizes the number of twists or loops in a cell's trajectory. It is given by the ratio of arclength to end-to-end distance between two points in a trajectory. We divide each trajectory into segments of arc-length $\approx 20\,\mu m$. We calculate the tortuosity for individual segments and find their mean for each trajectory. We consider the trajectories of all cells whose mean speed >1 μm/s and are imaged at 500 frames/s through ×40 objective for consistency. There were 52 H30 cells, 35 H10 Wobblers, and 23 H10 Synchronous cells which satisfied these conditions and the data from these cells constitute *Figure 2G*.

## Root mean square deviation

The match between experimental and theoretical flow fields is quantified by the RMSD of their velocities in the normalized scale ($v/v_{\max}$). $RMSD = \sqrt{\sum_{j=1}^{NG}(v_j^{\mathrm{expt}} - v_j^{\mathrm{th}})^2/NG}$, where $v_j^{\mathrm{expt}}$ and $v_j^{\mathrm{th}}$ are the experimental and theoretical values of the velocity fields at the $j$th grid point, respectively, and $NG$ is the total number of grid points. We calculate RMSD in the x and y components of the flow velocity, that is, in $v_x$ and $v_y$, respectively, for a comparison of the vector nature of the flow fields. This is because the signed magnitudes of $v_x$ and $v_y$ determine the vector direction of the flow. We also calculate RMSD in the flow speed ($|v| = [v_x^2 + v_y^2]^{1/2}$) to compare their scalar magnitudes.

## Acknowledgements

We acknowledge Aparna Baskaran, Ramin Golestanian, Ayantika Khanra, Swapnil J Kole, Malay Pal, Balachandra Suri, and Ronojoy Adhikari for useful discussions. This work is supported by the DBT/Wellcome Trust India Alliance Fellowship (grant number IA/I/16/1/502356) awarded to PS. SR acknowledges support from a J C Bose Fellowship of the SERB (India) and from the Tata Education and Development Trust.

## Additional information

### Funding

| Funder | Grant reference number | Author |
|---|---|---|
| The Wellcome Trust DBT India Alliance | IA/I/16/1/502356 | Prerna Sharma |
| Science and Engineering Research Board | J C Bose Fellowship | Sriram Ramaswamy |

The funders had no role in study design, data collection, and interpretation, or the decision to submit the work for publication.

### Author contributions

Debasmita Mondal, Conceptualization, Data curation, Formal analysis, Investigation, Methodology, Software, Validation, Visualization, Writing - original draft, Writing - review and editing; Ameya G Prabhune, Initial preliminary theoretical calculations; Sriram Ramaswamy, Conceptualization, Methodology, Project administration, Resources, Supervision, Writing - review and editing; Prerna Sharma,

Conceptualization, Funding acquisition, Methodology, Project administration, Resources, Supervision, Writing - original draft, Writing - review and editing

### Author ORCIDs
Debasmita Mondal http://orcid.org/0000-0002-8265-6876
Sriram Ramaswamy http://orcid.org/0000-0001-7726-8556
Prerna Sharma http://orcid.org/0000-0003-4988-9560

### Decision letter and Author response
Decision letter https://doi.org/10.7554/eLife.67663.sa1
Author response https://doi.org/10.7554/eLife.67663.sa2

## Additional files

### Supplementary files
• Transparent reporting form
• Source code 1. Custom-written MATLAB source codes.

### Data availability
All data generated or analyzed during this study are included in the manuscript and supporting files. Separate source data files containing source data for each subfigure have been provided. A source code file containing the custom-written MATLAB codes has also been provided.

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

## Appendix 1

### 1. Power dissipated through the flow fields

In low-Reynolds-number flows, the power $P$ generated by a microswimmer is dissipated through the induced flow fields as $P = 2\eta \int_V (\Gamma : \Gamma) \, dV$ (*Guasto et al., 2010*). Here, $\eta$ is the fluid viscosity, $\Gamma = \frac{1}{2}[\nabla v + (\nabla v)^T]$ is the fluid strain rate due to gradients in the flow velocity $v$, and the integral is over the quasi-2D chamber of height $H$. Roughly, for flows in bulk or in 2D fluid films, the velocity gradient along the chamber height is negligible and only the 2 × 2 part of $\Gamma$ corresponding to directions in the plane perpendicular to the confinement direction has non-negligible components (*Guasto et al., 2010*). This is not true in our case because the rigid boundaries act as momentum sinks, imposing a significant gradient in the fluid flow along the confinement direction $z$. Since the flow velocity varies as $v(x, y, z) = v^0(x, y) \cos(\pi z / H)$ (refer to *Figure 4—figure supplement 1* and associated main text), the norm-squared strain rate tensor for hard-wall confined flows is given by $\Gamma : \Gamma = (\Gamma : \Gamma)^{\text{bulk}} + \frac{(\pi v^0)^2}{2H^2} \sin^2\left(\frac{\pi z}{H}\right)$ where $(\Gamma : \Gamma)^{\text{bulk}} = (\partial_x v_x)^2 + \frac{1}{2}(\partial_y v_x + \partial_x v_y)^2 + (\partial_y v_y)^2$ and $v^0 = (v_x, v_y)$ is the flow profile in the swimmer's $x - y$ plane that is experimentally measured in *Figure 3*. We calculate the viscous power dissipation from the beat-averaged flow fields of CR to be $P^{30}$ =0.78 fW in weak confinement and $P^{10}$ =1.05 fW in strong confinement. These values are of the same order for both types of confinement and also to that measured for CR in thin fluid films ($P_{\text{mean flow}}$ in Figure 4a of *Guasto et al., 2010*).

### 2. Comparison of our experimental flow data in strong confinement with Liron and Mochon's theoretical solution

The far-field solution of Liron and Mochon for a parallel Stokeslet, $F$ located midway between two no-slip plates is given by $v_i^{LM}(r) = Q^{SD}\left(-\frac{\delta_{ij}}{r^2} + \frac{2r_i r_j}{r^4}\right)F_j$, which is equivalent to that of a 2D source dipole of strength $Q^{SD} = \frac{3H}{8\pi\eta} \frac{z}{H}\left(1 - \frac{z}{H}\right)$ (*Liron and Mochon, 1976*).

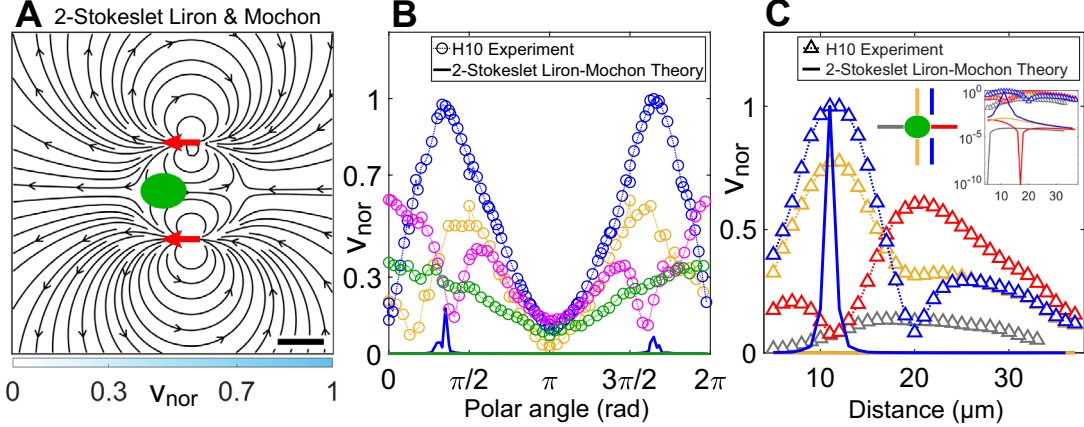

**Appendix 1—figure 1.** Theoretically computed flow field in confinement from Liron and Mochon's formula. (**A**) Theoretically computed flow field using Liron and Mochon's solution for two-Stokeslet model in 2D. The red arrows at (6, ±11) µm denote the position of the Stokeslets. The colourbar represents flow magnitude normalized by its maximum, $v_{nor}$. Scale bar, 10 µm. (**B**) Comparison between normalized experimental flow of a cell swimming in $H = 10\,\mu m$ (*Figure 3C*) and Liron and Mochon's theoretical flow field (**A**) along representative radial distances, $r$ from the cell centre as a function of polar angle; $r\,(\mu\text{m}) = 7$ (yellow), 13 (blue), 20 (magenta), and 30 (green). (**C**) Flow magnitude variation along four directions as indicated by separate colours in the *middle* inset (lateral to vortex [blue], lateral to cell centre [yellow], anterior [red], posterior [grey]) for the normalized experimental (symbols) and theoretical (solid lines) velocity fields in *Figure 3C* and (**A**), respectively. Except for the theoretical speed along the vortex direction (blue), others are negligible compared to the experiment as shown in the *rightmost* inset, which is a semilog plot of (**C**) in the $y$-axis.

As we have shown that the hydrodynamic cell drag is negligible to the flagellar thrust, the cell-body drag is insignificant and the observed flow field is mostly due to flagellar thrust. We,

therefore, superpose Liron and Mochon's solution for two flagellar forces and obtain the flow in *Appendix 1—figure 1A*. The streamlines of the 'two-Stokeslet Liron and Mochon flow' are qualitatively similar to that of the experiment (*Figure 3C*). However, the two-Stokeslet theoretical flow of Liron and Mochon decays much more rapidly than the experimental one and does not capture the experimental flow variation as shown in *Appendix 1—figure 1B,C*. Notably, there is no signature of vortex position lateral to the forcing point, that is, no minimum in the blue solid curve in *Appendix 1—figure 1C* because $v^{LM}$ is singular. Therefore, this far-field limit of the theoretical model is insufficient to describe the near-field flow variation, positions of vortices and other flow features of the strongly confined flow accurately. The root mean square deviation (RMSD) in $v_x$, $v_y$, and $|v|$ between the experimental flow of a H10 cell (*Figure 3C*) and two-Stokeslet Liron and Mochon's flow is 25.9%, 16.8%, and 30.8%, respectively (see Materials and methods for RMSD definition).

## 3. Inverse Fourier transform of the quasi-2D Brinkman equation in Fourier space

The quasi-2D Brinkman equation in Fourier space, *Equation 2* in the main text, is

$$v_k = \frac{O_k \cdot F}{\eta \left( k^2 + \frac{\pi^2}{H^2} \right)} \tag{A1}$$

Here, the orthogonal projection operator in polar coordinates $(k, \theta)$ is

$$O_k = 1 - \widehat{kk} = \begin{bmatrix} 1 - \widehat{k_x}^2 & -\widehat{k_x}\widehat{k_y} \\ -\widehat{k_y}\widehat{k_x} & 1 - \widehat{k_y}^2 \end{bmatrix} = \begin{bmatrix} \sin^2\theta & -\sin\theta\cos\theta \\ -\sin\theta\cos\theta & \cos^2\theta \end{bmatrix} \tag{A2}$$

where $\theta$ is the angle between wave vector $k$ and $x$-axis. For Stokeslets/Gaussian forces pointing along $x$- direction only, as in our case, $F = \begin{bmatrix} F \\ 0 \end{bmatrix}$, therefore $O(k) \cdot F = \begin{bmatrix} \sin^2\theta \\ -\sin\theta\cos\theta \end{bmatrix} F$.

To compute the velocity field in real space, we inverse Fourier transform *Equation A1* in polar coordinates, by replacing the numerator as shown above

$$v(r) = \frac{1}{(2\pi)^2\eta} \int e^{ik \cdot r} \begin{bmatrix} \sin^2\theta \\ -\sin\theta\cos\theta \end{bmatrix} \frac{F\, k\, dk\, d\theta}{\left( k^2 + \frac{\pi^2}{H^2} \right)} \tag{A3}$$

In polar coordinates, the field points in the $x - y$ plane are given by $(x, y) = (r\cos\phi, r\sin\phi)$, hence $k \cdot r = kr\cos(\theta - \phi)$. Thus, the fluid velocity field is

$$\begin{bmatrix} v_x \\ v_y \end{bmatrix}(r, \phi) = \frac{F}{4\pi^2\eta} \int_0^{2\pi} d\theta \int_0^\infty dk \begin{bmatrix} \sin^2\theta \\ -\sin\theta\cos\theta \end{bmatrix} \frac{ke^{ikr\cos(\theta - \phi)}}{\left( k^2 + \frac{\pi^2}{H^2} \right)} \tag{A4}$$

Let us change the $\theta$ integral from $(0, 2\pi) \to (-\pi/2 + \phi, \pi/2 + \phi)$, where $\cos(\theta - \phi) > 0$. For example, the $\theta$ integral for $v_x$ changes as follows,

$$\int_0^{2\pi} \sin^2\theta e^{ikr\cos(\theta - \phi)} d\theta = \int_{-\frac{\pi}{2}+\phi}^{\frac{\pi}{2}+\phi} \sin^2\theta e^{ikr\cos(\theta - \phi)} d\theta + \int_{\frac{\pi}{2}+\phi}^{\frac{3\pi}{2}+\phi} \sin^2\theta e^{ikr\cos(\theta - \phi)} d\theta \tag{A5}$$

Replacing $\theta \to \theta - \pi$ in the 2nd integral, the limits change as $(\pi/2 + \phi, 3\pi/2 + \phi) \to (-\pi/2 + \phi, \pi/2 + \phi)$, and the integrands $\sin\theta \to -\sin\theta$, $\cos\theta \to -\cos\theta$, $\cos(\theta - \phi) \to -\cos(\theta - \phi)$. Therefore, the second integral in the above equation changes to $\int_{-\frac{\pi}{2}+\phi}^{\frac{\pi}{2}+\phi} \sin^2\theta e^{-ikr\cos(\theta - \phi)} d\theta$. Hence, $v_x$'s $\theta$ integral becomes

$$\int_0^{2\pi} \sin^2\theta \, e^{ikr\cos(\theta-\phi)} d\theta = 2 \int_{-\frac{\pi}{2}+\phi}^{\frac{\pi}{2}+\phi} \sin^2\theta \cos[kr\cos(\theta-\phi)] d\theta \tag{A6}$$

Similarly, $\int_0^{2\pi} -\sin\theta\cos\theta \, e^{ikr\cos(\theta-\phi)} d\theta = 2\int_{-\frac{\pi}{2}+\phi}^{\frac{\pi}{2}+\phi} -\sin\theta\cos\theta\cos[kr\cos(\theta-\phi)]d\theta$. Thus, the velocity field in polar coordinates is given by,

$$\begin{bmatrix} v_x \\ v_y \end{bmatrix}(r,\phi) = \frac{F}{2\pi^2\eta} \int_{-\frac{\pi}{2}+\phi}^{\frac{\pi}{2}+\phi} d\theta \int_0^\infty dk \begin{bmatrix} \sin^2\theta \\ -\sin\theta\cos\theta \end{bmatrix} \frac{k\cos[kr\cos(\theta-\phi)]}{\left(k^2 + \frac{\pi^2}{H^2}\right)} \tag{A7}$$

For Gaussian forces, the numerator just gets multiplied by $e^{-k^2\sigma^2/2}$. We perform these 2D integrals in MATLAB for a 20 × 20 *xy* grid, with $k$ integral ranging from 0 to 100 to obtain the theoretical flow fields in this article.

The above integration takes 3 hr of computational time for two Stokeslets whereas it takes only 1 min to compute the flow field for 2 Gaussian forces of $\sigma = 5\,\mu m$ (*Processor:* Intel i7-4770 CPU with clock speed 3.4 GHz). Hence, we try to write a semi-analytical expression for the case of two Stokeslets. Let us consider $kr\cos(\theta-\phi) = p$ and $\frac{\pi r\cos(\theta-\phi)}{H} = q$. Then the $k-$ integral changes from $\int_0^\infty \frac{k\cos[kr\cos(\theta-\phi)]}{(k^2+\pi^2/H^2)} dk \to \int_0^\infty \frac{p\cos p}{p^2+q^2} dp$. We rename this integral as $I(q)$ and calculate it using the Exponential Integral, Ei (Equation 3.723—5 of *Gradshteyn and Ryzhik, 2007*).

$$I(q) = \int_0^\infty \frac{p\cos p}{p^2+q^2} dp = -\frac{1}{2}\left[e^{-q}\overline{\text{Ei}}(q) + e^q\,\text{Ei}(-q)\right] \tag{A8}$$

where,

$$\text{Ei}(q) = -\int_{-q}^\infty \frac{e^{-m}}{m} dm = \int_{-\infty}^q \frac{e^m}{m} dm, \quad \text{for } q < 0 \tag{A9}$$

and to avoid the singularity for $q > 0$, it is defined by using the principal value of the integral as

$$\overline{\text{Ei}}(q) = \int_{-\infty}^{-\epsilon} \frac{e^m}{m} dm + \int_\epsilon^q \frac{e^m}{m} dm, \text{ where } \epsilon > 0, \quad \text{for } q > 0 \tag{A10}$$

In our case $q > 0$, so we use *Equation A9* for calculating $\text{Ei}(-q)$ and *Equation A10* for calculating $\overline{\text{Ei}}(q)$, wherein we use $\epsilon = 10^{-5}$. So, *Equation A7* reduces to

$$\begin{bmatrix} v_x \\ v_y \end{bmatrix}(r,\phi) = \frac{F}{2\pi^2\eta} \int_{-\frac{\pi}{2}+\phi}^{\frac{\pi}{2}+\phi} d\theta \begin{bmatrix} \sin^2\theta \\ -\sin\theta\cos\theta \end{bmatrix} I(q) \tag{A11}$$

This method computes the flow field for two Stokeslets in 12 min on the same processor.

## 4. Swimmer-based Péclet number

Generally, speed and length scales in the definition of Péclet number are given by the swimmer speed, $u$, and radius, $R$ which we refer to as the swimmer-based Péclet number, $Pe_c = uR/D_S$. By this definition, $Pe_c^{30} \approx 0.6$ and $Pe_c^{10} \approx 0.02$ for the weakly and strongly confined CR, respectively. However, we note that the flow field closer to the cell surface is dominated by the vortices lateral to the cell body (*Figure 3A, C*), whose magnitude is significantly higher than the swimmer speed for the strongly confined cell ($V/u \sim 11$), in contrast to that of the weakly confined cell ($V/u \sim 0.3$). Hence, the flow-based Péclet number is more appropriate for describing the enhancement of mass transport of solutes due to the vortical flow fields generated by the flagella, particularly for the strongly confined cell ($H = 10\,\mu m$). This is shown below (*Appendix 1—table 1*) to be 100 times higher than the swimmer-based Péclet number, whereas both definitions yield almost similar $Pe$ for the weakly confined cell ($H = 30\,\mu m$).

**Appendix 1—table 1.** Flow-based Péclet number calculation from the flow fields.

| | $H = 30\,\mu m$ | $H = 10\,\mu m$ |
|---|---|---|
| Vortical flow speed, $V$ ($\mu$m/s) (*Figure 3A, C*) | 30 | 45 (also frontal flow) |
| Vortical diameter, $l_V$ ($\mu m$) 2 × vortex point distance (*Figure 3B, D*) | 2 × 8.5 = 17 | 2 × 20 = 40 |
| $t_{\mathrm{adv}} = l_V/V$ (s) | 0.57 | 0.8 |
| $t_{\mathrm{diff}} = l_V^2/D_S$ (s) | 0.3 | 1.6 |
| $Pe = t_{\mathrm{diff}}/t_{\mathrm{adv}} = l_V V/D_S$ | 0.5 | 2 |

## 5. Comparison of our theoretical model of strongly confined flow with that of *Jeanneret et al., 2019*

Jeanneret et al. provides an effective force-free 2D model for explaining the flow field of confined swimmers between 2 boundaries. They consider a force-free combination of 2D Brinkman Stokeslets along with a 2D source dipole to explain their experimental flows (*Jeanneret et al., 2019*). They use the analytical solution of *Pushkin and Bees, 2016* for their 2D Stokeslets with the permeability length $\lambda = H/\sqrt{12}$ (for the z-averaged flow in a Hele-Shaw cell of height *H*). They consider the conventional three-Stokeslet model of CR where the flagellar thrust, distributed between two Stokeslets of strength $-F_S/2$ each at $(x_1, \pm y_1)$, is balanced by the cell drag of strength $F_S$ at $(x_0, 0)$, all oriented along the direction of motion. Along with these force-free Stokeslets, they include the 2D source dipole of strength $l_d$ at $(x_d, 0)$. Finally, they used this model with six free parameters ($F_S$, $x_0$, $x_1$, $y_1$, $l_d$, and $x_d$) to fit their experimentally observed flow fields of CR in confinements ranging from 14 to 60 μm.

However, our theoretical model consists of a 2D Brinkman Stokeslet because the strongly confined CR exerts a net force on the fluid due to the presence of strong non-hydrodynamic contact friction from the walls, unlike that of *Jeanneret et al., 2019*. This force-monopole is spatially distributed equally at the two flagellar positions, each with a Gaussian regularization to describe the strongly confined flow due to the H10 cell. The reason our theoretical approach is not the same as *Jeanneret et al., 2019* is because there are two major differences in our experimental observations. First, we observe that the strongly confined H10 flow is mostly due to the flagellar motion with a 96% reduction in the cell's swimming speed, thanks to the static friction from the walls (compared to H30 cells), leading to the hydrodynamic cell-drag being nearly absent. This coupling between motility and confinement is not observed by *Jeanneret et al., 2019*, likely due to the slightly weak confinement ($D/H \lesssim 0.7$) produced by their experimental methodology, where the stresses present in the system are mostly hydrodynamic. It is therefore appropriate for them to use the force-free three-Stokeslet theoretical model for CR (apart from the source dipole contribution) whereas in our case, the nearly absent hydrodynamic drag experienced by the cell body leads to a monopolar flow with only two Stokeslets (like-signed) localized with a Gaussian spread around the approximate flagellar positions. Second, the spinning motion of CR cells is restricted in our strongly confined H10 chambers unlike those in *Jeanneret et al., 2019*. They added the extra 2D source dipole in their theoretical model to account for both finite-sized effects of the cell body and spinning motion of the cells [explained in Figure 1c of *Jeanneret et al., 2019*].

## 6. Is the two-Gaussian Brinkman model applicable to a collection of strongly confined pullers?

We analyze the fluid flow due to two strongly confined H10 Synchronous cells as a preliminary test for determining the applicability of our theoretical methodology to a collection of microswimmers. Specifically, we measure the beat-averaged flow field of two synchronously beating cells which are separated by $\sim 9$ body diameters and approach each other head-on (*Appendix 1—figure 2A*). Therefore, we linearly superpose the solution of the quasi-2D Brinkman equation for a pair of two-Gaussian forces ($\sigma = 5\,\mu m$) at the approximate flagellar positions of the two cells and obtain the resultant flow field (*Appendix 1—figure 2B*). The position and direction of flow vortices along with the stagnation point in between the two cells match well between the experiment and theory. This

suggests that linearly superposing two-Gaussian Brinkman flows might be an adequate description for the flow field of a dilute collection of CRs.

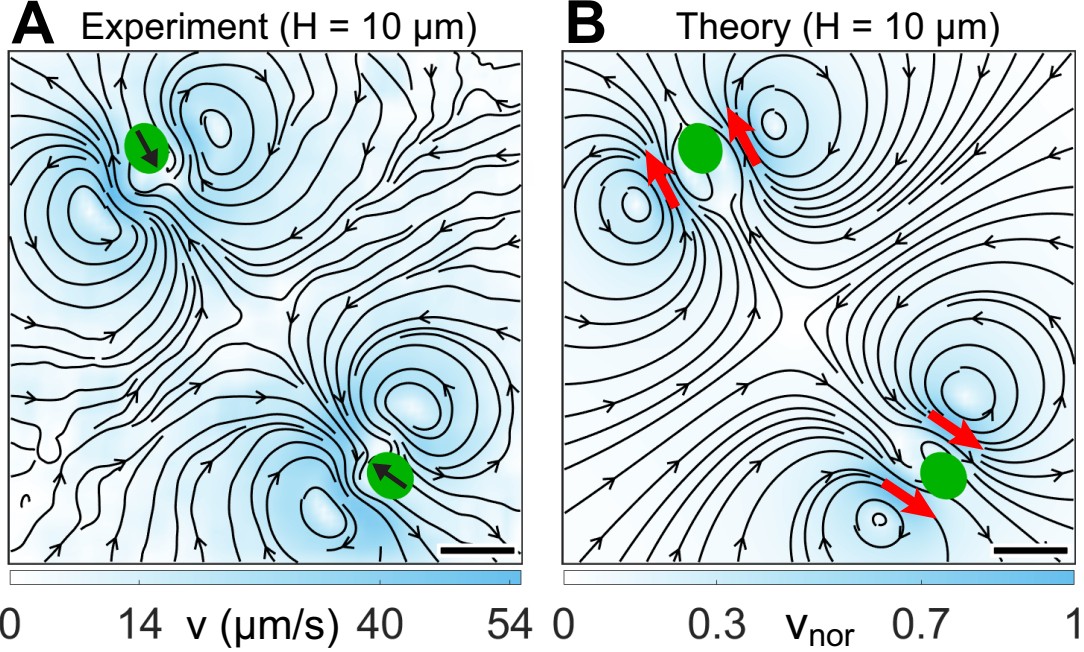

**Appendix 1—figure 2.** Flow fields due to two strongly confined H10 cells. (**A**) Experimentally measured flow field for two synchronous cells swimming in $H = 10\,\mu m$. This flow is averaged over ~30 beat cycles for each cell during which the cells move merely 0.05 times their respective body diameters ($D \sim 12.42\,\mu m$). The centre-to-centre distance between the swimmers is 8.75$D$. Black arrows on the cell bodies indicate their swimming direction. Solid black lines indicate the streamlines of the flow in lab frame. The colourbar represents flow magnitude, $v$. (**B**) Theoretically computed flow field by linearly superposing two two-Gaussian Brinkman flow, one for each cell. The positions of the pair of two-Gaussian forces at approximate flagellar positions are denoted by red arrows. The streamlines, vortex flows, and stagnation point at the centre of the grid match qualitatively with the experimental one (**A**). The colourbar represents flow magnitudes normalized by its maximum, $v_{nor}$. Scale bars, 20 μm.

