## [Editor Report]

The manuscript focusses on changes in the flow fields generated by swimming microorganisms as a consequence of them being squeezed within a very narrow gap. The resulting friction with the cell body is such that the flows are dominated by the propelling appendages only, and the authors show in this regime the surprising result that there can be an enhanced nutrient flux to the microorganism.

---

## [Decision Letter]

**Decision letter after peer review:**

Thank you for submitting your article "Strong confinement of active microalgae leads to inversion of vortex flow and enhanced transport" for consideration by *eLife*. Your article has been reviewed by 2 peer reviewers, and the evaluation has been overseen by a Reviewing Editor and Anna Akhmanova as the Senior Editor. The reviewers have opted to remain anonymous.

Essential revisions:

1) A general conclusion of the reviewers is that while they appreciate the approach taken to the Brinkman equation, reference should be made to other recent work in this area, including Pushkin and Bees (10.1007/978-3-319-32189-9_12), Jeanneret et al., (PRL; 10.1103/PhysRevLett.123.248102) and Fortune et al., (JFM; 10.1017/jfm.2020.1112). In addition, the revision should not only discuss previous work but also improve the interpretation of the results. The reviewers felt that the discussion should avoid mystifying the physics, which they believe is straightforward – when puller cells are subject to wall friction due to being in a tight space, they exert a net force towards themselves, and the associated flow field decays more slowly.

2) *Chlamydomonas* is not neutrally-buoyant (page 4).

3) The Peclet number calculation on page 10 is unclear. The diffusion coefficient is an order of magnitude estimate, whereas the Peclet number is calculated to one significant figure, and value of 2 versus 0.5 is considered a change of regime. Really all this shows is that diffusion and advection are both likely to be important for small molecules. Considering the effect of flow oscillations (order of magnitude estimate 10 microns x 250 rad/s = 2500 microns/s) suggests advection is more important still.

4) More data on beat frequency and how it varies with confinement would be useful. The impression is given that maybe it is near-constant?

5) Some of the presentation is unclear, for example the azimuthal variation of the flow fields (Figure 5BD and some of the appendix figures) it is difficult to see the scale and the variation is very compressed.

6) The argument for wall contact force is essentially correct I think but is unclearly written. If there is no contact force, the correct force balance taken over the cell F_drag + F_propulsive = 0. The wall changes the drag and propulsive force, but does not directly exert a force on the cell. If there is contact, then the balance is F_drag + F_propulsive + F_contact = 0. The last two sentences about 'could in principle originate in direct frictional contact' is confusing because this is actually the conclusion of the argument, not just something 'in principle'.

7) There is a lack of clarity about measures of variation versus measures of uncertainty. For example, the 'error bars' in Figure 2G represent standard deviation in a heterogeneous population, not a measure of experimental uncertainty.

8) Line 48. Should it be "primed" instead of "prime"?

9) Line 51. "[…]ecological characteristics and theoretical description". Unclear meaning

10) Line 53 (and in the discussion). "soft PDMS". I do not think that the Youngs modulus of the PDMS in a normal microfluidic device (which by the way is several mm thick) is small enough for the channels to be bent to any significantly degree by swimming microorganisms. I have many years of experience in this field and personally I have never seen any such thing.

11) Line 60,61. "We find that the cell speed decreases significantly and the trajectory tortuosity increases". Have the author checked whether -and how much- the rotational diffusivity of the cells change in confinement? Is the increased tortuosity of the trajectory just a consequence of their smaller speed?

12) Lines 65,66. "but also to those predicted from the source dipole theory of strongly confined swimmers"

Line 339-342. "This result is contrary to the common theoretical expectation that the far-field flow of a confined microswimmer between two closely spaced solid walls is a 2D source dipole pointing along the swimmer's propulsion direction". I find these sentences misleading. The "theoretical expectations" were made with the assumption that the only stresses present in the system are hydrodynamic. Therefore it is not surprising that one finds something different when this is not the case. The statements in the paper suggest to the reader that the previous theoretical models were wrong, whereas it is just that now the cells' friction with the walls needs to be taken into account.

13) I am not convinced by that the current method used to extract the friction force for the body. The authors rely on an estimate of the drag based on the bulk drag coefficient, which is certainly an underestimate of the hydrodynamic drag of the body under confinement. I agree that in the 10um case the cells move at such a low speed that the contribution from hydrodynamic friction is much smaller than that from contact friction with the wall. But at this point one might as well just forget about hydrodynamic friction rather than removing the wrong estimate for it.

14) Line 245. It would help the reader to include that v_rad^0 is equal to \vec{v}\dot\hat{r}.

*Reviewer #1:*

The manuscript aims to study (through particle tracking microscopy), understand (through mathematical modelling) and interpret (through analysis of mixing) the time-averaged flow fields around swimming algae when the cells are squeezed between rigid glass surfaces.

The main strengths of the work are the acquisition and analysis of extensive flow data, and development of a rather simple and elegant mathematical model for the flow, based on depth-averaging the viscous flow equations with spatially-smoothed force terms, and subsequent Fourier solution. This approach is much simpler than known solutions of the 3D flow equations involving infinite sequences of images or Hankel transforms, and/or computational solutions of the flow problem which resolve the cell body and flagella.

Weaknesses include:

a) The terminology around 'extreme/strong/weak confinement' (borrowing from quantum physics?) perhaps gives the impression of physical complexity, whereas what has been done is quite simple – cells were examined in a chamber in which there was quite a lot of space to move, or not very much space, or they were squeezed very tightly and barely able to move. In the first two cases they could move effectively through the fluid and so cell tends to drag fluid along from behind and push it forward in front. In the squeezed case, the cell is barely moving and so the flagella pull fluid from front to back, reversing the sense of the surrounding vortices.

b) Along similar lines, perhaps more is made of the use of 'hard versus soft' boundaries. In appendix 1.1 is it claimed that soft boundaries (such as PDMS?) will not produce a significant velocity gradient. However, the correct boundary condition on a soft solid surface is still the no-slip condition. I would think the important issue is the deformability of the boundary compared with the cell, and hence the level of friction resulting from squeezing.

c) The model is good, but the reasons for its success relative to the solution of Liron and Mochon that it is compared to are perhaps simpler than suggested. The approximation for the flow field due to a force monopole in a confined domain of Liron and Mochon (which is potential-dipole in character) is (i) singular, which is responsible for the spike in appendix figure 1c, (ii) in any case is the far-field limit of the full singular solution, so cannot be expected to be accurate in the near-field. Conversely, it is relatively unsurprising that the near-field of the cilia motion is better modelled by a spatially-averaged force. Whether the force needs to be Gaussian, or some other regularization, is unclear.

d) Both the data and the model are for time-averaged flow field, which loses the (large) oscillations occurring in the near-field flow due to the flagellar beat. These oscillations may have a role in mixing. Resolving this flow is probably very difficult experimentally, but can be accomplished in silico through computational modelling. The difference between time-averaged and instantaneous flow should at least be acknowledged.

e) The slower decay of the velocity correlation in the confined case (figure 5 – supplement figure 2) could be explained by the fact that the confined case produces a force monopole, and hence a lower order of decay than a force-free swimmer? In which case we are not necessarily seeing evidence of increased mixing?

f) Apart from a brief reference to a review paper, relatively little contact is made with *Chlamydomonas* biology or ecology, particularly concerning the functional importance of fluid mixing. I am not a specialist in this area; my question is, under what circumstances is the ability to exchange e.g. carbon dioxide with the surrounding fluid a limiting factor in metabolism? Are we seeing something specific to CR's adaptation to certain natural habitats, or just that tethered swimmers always disturb more fluid than free swimmers due to the force monopole?

Appraisal: I believe the authors have partially achieved their aim of studying and understanding the flow fields produced by confined swimming algae.

Likely impact: I believe the paper may help to clarify understanding of the fact that the flow field changes when a swimming cell is in contact with a surface; the mathematical approach will be valuable in producing simplified mechanistic models of the flow fields produced by cells in the ubiquitous microscope slide-coverslip set up.

*Reviewer #2:*

The manuscript focusses on changes in the flow fields generated by swimming microorganisms as a consequence of them being squeezed within a gap that is narrower than their size. This is studied here in the context of the unicellular green microalga *Chlamydomonas reinhardtii*, which is a common model system for microbial motility of body sizes ranging from 8um to 14um. The authors compare their behaviour in the two cases of a 30um and 10um-thick Hele-Shaw cell. The resulting flow fields are then compared with those obtained by a superposition of quasi-2D Stokeslets, Green functions of the Brinkman equation.

The paper has three main results. Firstly, cell behaviour in the 10um-thick samples depends clearly on microorganismal size. Larger cells beat their flagella with the standard synchronous breaststroke, while smaller ones display either asynchronous beating or "paddling" (as the authors call it, but see below my comment on this). Secondly, the larger cells display an average flow field that, in the far field, is directed oppositely to what would be predicted in absence of cell squeezing by the walls. Thirdly, the paper presents a theoretical modelling for the observed flow fields in terms of a "Gaussian" Brinkman Stokeslets. This flow field is proposed to increase the flow of nutrients to the cell.

1) The confinement that the authors focus on is in a different regime that what has been addressed earlier, in the sense that it starts to probe strong non-hydrodynamic friction, which is a case that can definitely happen in nature. This is interesting, although likely to depend quantitatively very much on topographical and chemical details of the actual surfaces that are in close contact.

Given that the strongly confined cells experience a non-hydrodynamic friction that is large enough to almost halt their swimming, it is not surprising that the measured flow field is dominated by the forces imposed on the fluid by the flagella and therefore a force-free 3-Stokeslet model is inappropriate. After all, the presence of non-hydrodynamic friction means that the cell exerts a net force on the fluid. It seems to me that this situation essentially resembles that of a tethered microorganism like Vorticella and I would have liked to see more discussion on the similarities between the two cases, both experimentally and from a modelling perspective. I do not think this is developed sufficiently in the manuscript.

Besides this, the range of gaps probed by the authors is rather coarse. The 10um gap is not much smaller than the smallest gaps that were probed in ref [12] for which -however- the flow field was qualitatively different. In my opinion, within the context of the effect of confinement on flow fields, it would have been interesting to explore this range in more detail, studying the transition from the hydrodynamic to the contact-friction case.

2) Regarding the modelling, I have two main comments. Firstly, I appreciate the idea of a diffused Stokeslet. However, it does look very close to the idea of a regularised Stokeslet (Cortez 2005), which is not even mentioned in the manuscript. I think this is quite surprising. I would expect the manuscript to comment on differences between the two approaches.

Secondly, I do not understand the FT approach to finding the 2D Stokeslet for the Brinkman equation, when this is actually known analytically. It has been published in

Pushkin, D. O. and Bees, M. A. Bugs on a slippery plane: Understanding the motility of microbial pathogens with mathematical modelling. Adv. Exp. Med. Biol. 915, 193-205 (2016).

which is cited in ref [12] of the manuscript (Jeanneret et al., PRL 2019). This paper also shows that a model based on the point forces from Liron and Mochon does not perform well (see Suppl Mat), while one made of Brinkman Stokeslets with a 2D source dipole does. Therefore I do not think that it is surprising to see that the same approach works well for the current manuscript. In fact, given the similarity between the current setup and that in [12], I would have expected a comparison between the current approach and the full model from [12] at least in the Supplementary Material.

Finally, I think that comparing the magnitude of the flow fields as in Figure 5B,D is insufficient. One should instead show that both magnitude and direction of the flow fields are well captured by the model. This cannot really be grasped by comparing by eye Figure 3C and Figure 5A, C.

3) As for the question of nutrient transport, I find the claim on the enhancement for the 10um-cells not completely convincing. It is clear that the fluid fluxes for the 30um-cells and the 10um-cells are organised in an opposite way. However, Figure 4D shows that the positive part of the flux is of a very similar magnitude in both cases, in particular for the average fluxes. This is true also when one compares the negative fluxes. Given the similarities, I do not find a compelling reason why the cell should have a much higher nutrient uptake in one of the two cases. This might be the case, but there is not enough evidence in the paper to support this statement.

4) Finally, regarding the “paddling” state, I’m afraid this is the normal flagellar shock response in *Chlamydomonas* and not some sort of previously unknown state of the cells. It is known that the shock response can be elicited both by intense light stimuli or by mechanical stimuli. It is very likely that this is just a mechanosensitive shock, which comes from the mechanical interactions between the flagella and the upper/lower walls. In turn, these are possible due to the fact that cells of smaller size than the gap can spin around their body enough to touch the walls with their flagella. I think that the authors would need to investigate further the existing literature on flagellar shock response in *Chlamydomonas* and put appropriately in context the “paddling” behaviour they observe.

---

## [Author Response]

Essential revisions:1) A general conclusion of the reviewers is that while they appreciate the approach taken to the Brinkman equation, reference should be made to other recent work in this area, including Pushkin and Bees (10.1007/978-3-319-32189-9_12), Jeanneret et al., (PRL; 10.1103/PhysRevLett.123.248102) and Fortune et al., (JFM; 10.1017/jfm.2020.1112). In addition, the revision should not only discuss previous work but also improve the interpretation of the results. The reviewers felt that the discussion should avoid mystifying the physics, which they believe is straightforward – when puller cells are subject to wall friction due to being in a tight space, they exert a net force towards themselves, and the associated flow field decays more slowly.

We thank the editor for summarising the reviewers’ comments and suggesting the changes to the discussion. We now cite recent related works and discuss their connection to our approach in the revised manuscript. We’re sorry our text gave the impression of “mystifying” a simple but worthwhile piece of physics, and we hope our modified discussion is free of this defect.

2) Chlamydomonas is not neutrally-buoyant (page 4).

We thank the editor for pointing this out. Indeed, *Chlamydomonas reinhardtii* (CR) is not neutrally buoyant, rather its density is 5% greater than that of water [Drescher et al., 2010]. However, CR can still be considered as a force-free swimmer because the sedimentation speed is far smaller than its swimming velocity [Ishikawa et al., 2006, Mathijssen et al., 2015]. The gravitational effect due to the excess density (Δρ∼50 kg/m3) results in the sedimentation speed to be given by us=Fg6πηR=2ΔρR2g9η where Fg is the buoyant weight of CR, R≈5μm is the cell body radius, g is the acceleration due to gravity and η=1mPa.s is the viscosity of the medium. Therefore, us∼2.72 μm/s which is about 2% of the swimming speed u30∼120 μm/s of CR in bulk. Hence, the Stokeslet component in the flow due to Fg is negligible for the near-field distances r<35R [Drescher et al., 2010] and it is appropriate to consider the swimmer as force-free over the distances where we are measuring the flow field.

We remove the word ‘neutrally buoyant’ and rephrase the statement at the end of page 4 of the revised draft as:

“The net force and torque on microswimmers, together with the ambient medium and boundaries, can be taken to be zero as gravitational effects are negligible in the case of CR for the range of length scales considered [Drescher et al., 2010].”

3) The Peclet number calculation on page 10 is unclear. The diffusion coefficient is an order of magnitude estimate, whereas the Peclet number is calculated to one significant figure, and value of 2 versus 0.5 is considered a change of regime. Really all this shows is that diffusion and advection are both likely to be important for small molecules. Considering the effect of flow oscillations (order of magnitude estimate 10 microns x 250 rad/s = 2500 microns/s) suggests advection is more important still.

We appreciate the editor’s viewpoint and agree that our Peclet number calculation only suggests that both diffusion and advection are important for most of the biological molecules (nutrient salts, oxygen etc.) which are small. It is evident from the cell suspension videos that the tracers are advected more by the H10 cell than the H30 cell (Video 1 and 3). Hence, as a first and standard measure to characterize the relative significance of advective to diffusive transport, we calculate the Peclet number. We agree with the editor that we should not consider the value of 2 vs 0.5 as a change of regime and that all it says that both advection and diffusion are important for the confined CR cells.

We also agree that the order-of-magnitude estimates of flagella-driven flow oscillations (vosc) suggest that advection is important for both the H30 (νb∼55 Hz, vosc∼3450 μm/s) and H10 (νb∼52 Hz, vosc∼3270 μm/s) cells as their beat frequencies (νb) are similar. However, we are interested in the long-time behaviour where these flow oscillations are averaged out and the recorded videos of the H10 cell suspensions hint at enhanced advection due to beat-averaged flows. The slower decay of the velocity correlation for the strongly confined H10 flow (revised *Figure 5A*) already supports this observation. In addition, we now include direct evidence of enhanced mixing and transport of tracers due to the H10 flows (revised *Figure 5, B and C; Figure 5 —figure supplement 1*), thanks to the suggestion of the reviewers (please see our response to point (e) of Reviewer 1). This explicitly shows that the fluid is indeed advected more in the strongly confined case than the weakly confined one. Calculation of the Peclet number was simply a preamble at attempting to quantify the increased fluid transport and mixing through flow correlation length scales and mean-squared displacement of tracers averaged over several flagellar beat periods.

We acknowledge the editor’s suggestions by modifying the text related to Peclet number in our revised manuscript appropriately (pg. 18). We also add a few lines acknowledging the importance of flow oscillations in fluid mixing as suggested by the editor in the discussion of our revised manuscript (pg. 21).

4) More data on beat frequency and how it varies with confinement would be useful. The impression is given that maybe it is near-constant?

We thank the editor for this relevant suggestion. We measure the beat frequency of the strongly confined H10 Synchronous cells and weakly confined H30 cells (when their flagellar beat is in the image plane, the same data that is considered for analyzing the flow fields). The flagellar beat pattern of the H10 Wobblers is irregular and hence we cannot assign a beat period to these cells.

The beat frequency of H10 Synchronous cells (degree of confinement, DH∼1.2) is νb10≈51.58±7.62 Hz (averaged over 210 beat cycles from 20 representative cells) and that of H30 cells (DH∼0.35) is νb30≈55.27±8.22 Hz (averaged over 194 beat cycles from 20 representative cells). Therefore, the flagellar beat frequency is similar with varying confinement. This is because even in the 10 μm chamber where the CR cell body is strongly confined, the flagella are beating far from the walls (∼ 5 μm) and almost unaffected by the confinement.

We acknowledge the editor’s suggestion by including this data in our revised manuscript (page 8).

5) Some of the presentation is unclear, for example the azimuthal variation of the flow fields (Figure 5BD and some of the appendix figures) it is difficult to see the scale and the variation is very compressed.

We appreciate the editor’s concern and acknowledge it by modifying all the comparison between the azimuthal variation of the experimental and theoretical flow fields to the same normalised scale between 0 and 1, without any compression (see revised Figures 4B, 4D, Figure 4—figure supplement2, Figure 4—figure supplement3, Appendix1-Figure1B). In these plots, we have included the azimuthal variation for four representative radial distances from the cell centre, r = 7, 13, 20 and 30 μm, for clarity.

6) The argument for wall contact force is essentially correct I think but is unclearly written. If there is no contact force, the correct force balance taken over the cell F_drag + F_propulsive = 0. The wall changes the drag and propulsive force, but does not directly exert a force on the cell. If there is contact, then the balance is F_drag + F_propulsive + F_contact = 0. The last two sentences about 'could in principle originate in direct frictional contact' is confusing because this is actually the conclusion of the argument, not just something 'in principle'.

We appreciate the suggestion of the editor. We completely agree as regards the force balance equation in the absence and presence of contact force. We remove the sentence “The forces exerted by the cell…” while introducing the wall drag at the beginning of the section – Force balance on confined cells. We rephrase and re-write this whole section of force balance (pg. 12) to clearly state the contribution from different forces in our revised manuscript and it acknowledges the editor’s suggestion in the last sentence:

“Thus the flagellar thrust works mainly against the non-hydrodynamic contact friction from the walls as expected due to the extremely low speed of the strongly confined swimmer.”

7) There is a lack of clarity about measures of variation versus measures of uncertainty. For example, the 'error bars' in Figure 2G represent standard deviation in a heterogeneous population, not a measure of experimental uncertainty.

We agree with the editor and accordingly, clarify the statement in Figure 2G as:

“The error bars in the plot corresponds to standard deviation *due to the heterogenous population of cells*.”

8) Line 48. Should it be "primed" instead of "prime"?

We thank the editor for seeking clarification. We have now modified the word ‘stress-prime’ for better readability in the revised manuscript (line 64):

“…extreme confinement between two hard walls has been exploited to *induce stress memory in* CR cells towards enhanced biomass production and cell viability.”

9) Line 51. "[…]ecological characteristics and theoretical description". Unclear meaning

We replace the word ‘ecological characteristics’ with ‘fluid flow and mixing’ in the revised manuscript and the modified sentence in line 68 reads as “…how rigid walls might modify the kinetics, kinematics, fluid flow and mixing, and theoretical description of a strongly confined microalga such as CR is scarce …”

10) Line 53 (and in the discussion). "soft PDMS". I do not think that the Youngs modulus of the PDMS in a normal microfluidic device (which by the way is several mm thick) is small enough for the channels to be bent to any significantly degree by swimming microorganisms. I have many years of experience in this field and personally I have never seen any such thing.

We thank the editor for pointing this out. We agree with the editor that the elastic modulus of PDMS is too high for the microorganism to bend the microfluidic device. We, therefore, refrain from using the word ‘soft’ for PDMS chambers in the introduction and discussion of the revised manuscript.

11) Line 60,61. "We find that the cell speed decreases significantly and the trajectory tortuosity increases". Have the author checked whether -and how much- the rotational diffusivity of the cells change in confinement?

We appreciate the editor for raising this concern and seeking clarifications. We cannot measure the rotational diffusivity of the cells from the data we have used in Figure 2G (cell speed and tortuosity) because the trajectories are captured through 40X objective and hence, we have access to only short-time data. Therefore, we capture long-time trajectories of CR cells in chambers of height 10 and 30 μm through a 10X bright-field objective. The only drawback of these data is that we cannot differentiate Wobblers from Synchronous cells in H=10 μm robustly as we cannot observe their flagellar beat through bright-field microscopy. We filter the trajectories whose instantaneous speed < 0.5 μm/s to remove the cells which are stuck to the glass surfaces at some points in their trajectory through flagella or cell body, at the cost of excluding some Synchronous (non-stuck) cells from the analyses because their speed is similar to that of a stuck cell.

The rotational diffusion coefficients, DR, extracted from the mean squared angular displacement vs lag time plot (from 10X data) for the H30 cells are DR30∼0.11 rad^2^/s and that of the H10 cells are DR10∼0.76 rad^2^/s. That is, DR increases by almost 85% when the confinement increases from 30 μm to 10 μm. The corresponding increase in tortuosity (from 40X data shown in Figure 2G) from H30 to H10 cells is approximately 60%. Therefore, the increase in tortuosity with increasing confinement is correlated with increase in the rotational diffusivity of cells.

Is the increased tortuosity of the trajectory just a consequence of their smaller speed?

The increase in tortuosity is not because of the decrease in speed with increasing confinement. It depends on the nature of the walk i.e., the statistics of angular displacement, rather than the speed magnitude. In principle, a slow swimmer can still move along straight lines without frequent turning events which will lead to a greater persistence and thereby low tortuosity in the trajectories (as well as low rotational diffusivity of the cells).

12) Lines 65,66. "but also to those predicted from the source dipole theory of strongly confined swimmers"Line 339-342. "This result is contrary to the common theoretical expectation that the far-field flow of a confined microswimmer between two closely spaced solid walls is a 2D source dipole pointing along the swimmer's propulsion direction". I find these sentences misleading. The "theoretical expectations" were made with the assumption that the only stresses present in the system are hydrodynamic. Therefore it is not surprising that one finds something different when this is not the case. The statements in the paper suggest to the reader that the previous theoretical models were wrong, whereas it is just that now the cells' friction with the walls needs to be taken into account.

We thank the editor for pointing this out. We modify these sentences in the text of our revised manuscript to clearly state the conditions in which previous theoretical models are applicable.

13) I am not convinced by that the current method used to extract the friction force for the body. The authors rely on an estimate of the drag based on the bulk drag coefficient, which is certainly an underestimate of the hydrodynamic drag of the body under confinement. I agree that in the 10um case the cells move at such a low speed that the contribution from hydrodynamic friction is much smaller than that from contact friction with the wall. But at this point one might as well just forget about hydrodynamic friction rather than removing the wrong estimate for it.

We thank the editor for raising this concern. We agree that the contribution from the zeroth-order Stokes drag is an underestimate of the net hydrodynamic drag of the cell body under strong confinement. We, therefore, remove this estimate as the total hydrodynamic cell drag for the strongly confined cells in the revised manuscript and re-write the section ‘Force balance on confined cells’ (pg. 12, purple coloured text) to acknowledge this concern.

14) Line 245. It would help the reader to include that v_rad^0 is equal to \vec{v}\dot\hat{r}.

We thank the editor for this suggestion. In the revised manuscript, we would have included vrad0(r,ϕ)=−v0⋅r^=−vxcosϕ−vysinϕ which denotes the *inwards* radial flow in the z=0 plane i.e., the flow towards the cell situated at the origin of the coordinate system and hence the negative sign. However, we have replaced this fluid flux analysis with direct evidence of enhanced mixing in revised Figure 5 and associated text in response to the suggestion (3) of reviewer 2. Therefore, this text is not present is the revised manuscript.

Reviewer #1:The manuscript aims to study (through particle tracking microscopy), understand (through mathematical modelling) and interpret (through analysis of mixing) the time-averaged flow fields around swimming algae when the cells are squeezed between rigid glass surfaces.The main strengths of the work are the acquisition and analysis of extensive flow data, and development of a rather simple and elegant mathematical model for the flow, based on depth-averaging the viscous flow equations with spatially-smoothed force terms, and subsequent Fourier solution. This approach is much simpler than known solutions of the 3D flow equations involving infinite sequences of images or Hankel transforms, and/or computational solutions of the flow problem which resolve the cell body and flagella.

We sincerely thank the reviewer for appreciating our work and providing us positive feedback.

Weaknesses include:a) The terminology around 'extreme/strong/weak confinement' (borrowing from quantum physics?) perhaps gives the impression of physical complexity, whereas what has been done is quite simple – cells were examined in a chamber in which there was quite a lot of space to move, or not very much space, or they were squeezed very tightly and barely able to move. In the first two cases they could move effectively through the fluid and so cell tends to drag fluid along from behind and push it forward in front. In the squeezed case, the cell is barely moving and so the flagella pull fluid from front to back, reversing the sense of the surrounding vortices.

We thank the reviewer for raising this concern. Our use of the terminology wasn’t inspired by quantum physics but rather relied on the fact that these terms are commonly used in the existing literature concerning microswimmers. For example, Brotto et al., 2013; Mathijssen et al., 2016 and Jeanneret et al., 2019 frequently use terms such as “strongly confined geometries”, “strong confinement”, “weak confinement” in the main text to describe the experimental/theoretical analysis conditions for microswimmers. Hence, in accordance with the existing literature we use the term ‘strong/extreme’ confinement when the ratio of average diameter of cells to chamber height, DH∼1.2 and ‘weak’ confinement when DH∼0.3. We would like to re-draw the attention of the reviewer to the existing statement in our manuscript (pg. 8) “Henceforth, we equivalently refer to the H10 Synchronous CR as ‘strongly confined’ or H10 cells (DH≳1) and the H30 cells as ‘weakly confined' (DH<1).” We feel this sentence defines the terminology explicitly for the reader’s convenience and addresses the concern sufficiently well.

b) Along similar lines, perhaps more is made of the use of 'hard versus soft' boundaries. In appendix 1.1 is it claimed that soft boundaries (such as PDMS?) will not produce a significant velocity gradient. However, the correct boundary condition on a soft solid surface is still the no-slip condition. I would think the important issue is the deformability of the boundary compared with the cell, and hence the level of friction resulting from squeezing.

We thank the reviewer for pointing this out. Indeed, any microfluidic chamber with a solid surface will induce a no-slip boundary condition. However, in appendix 1.1, we mean that the soft boundaries are that due to a freely-suspended fluid film such as a soap film which has stress-free film boundaries at the fluid-air interface (e.g., the experiments of [Guasto et al., 2010; Wu and Libchaber, 2000]) and not a chamber made of PDMS. In such a quasi-2D fluid film with no solid boundaries, there is no velocity gradient along the height of the film (Supplementary materials of [Guasto et al., 2010]). But we completely agree with the reviewer that the elastic modulus of PDMS is too high for the microorganism to deform the microfluidic device. We, therefore, use the term ‘soft’ carefully and modify ‘soft 2D film’ to ‘thin fluid film’ in appendix 1.1 as well as refrain from using the word ‘soft’ for PDMS chambers in introduction and discussion.

c) The model is good, but the reasons for its success relative to the solution of Liron and Mochon that it is compared to are perhaps simpler than suggested. The approximation for the flow field due to a force monopole in a confined domain of Liron and Mochon (which is potential-dipole in character) is (i) singular, which is responsible for the spike in appendix figure 1c, (ii) in any case is the far-field limit of the full singular solution, so cannot be expected to be accurate in the near-field. Conversely, it is relatively unsurprising that the near-field of the cilia motion is better modelled by a spatially-averaged force. Whether the force needs to be Gaussian, or some other regularization, is unclear.

We thank the reviewer for this pertinent suggestion. We agree that the solution of Liron and Mochon cannot be expected to match our experimental flow data in the near-field because of the reasons (i) and (ii) mentioned above. We have now added and modified texts to acknowledge these reasons in our revised manuscript (see the first paragraph of section ‘Theoretical model of strongly confined flow’ and Appendix 1.2).

We agree with the reviewer that it may be relatively unsurprising that the near-field flow characteristics due to the flagellar motion are better modelled by a spatially averaged force. Unfortunately, the evidence of using such regularization for comparison with the experimentally observed flagellar flows is elusive in the literature. Mostly, the very near-field flow is theoretically described using a line distribution of Stokeslets along the flagella, which also works well. But, on the spatial scale that we are observing the flow, use of a single Gaussian force is a neat trick to explain the coarse-grained flagellar flow easily.

We now acknowledge the method of regularized Stokeslets [Cortez, 2001; Cortez et al., 2005] which is in a manner similar to our approach, thanks to the suggestion of Reviewer 2. Our convolution approach differs from this method in its implementation – our method gives numerical solution to any form of regularization to the Stokeslet whereas Cortez et al., approach gives analytical closed-form solutions to the same. The reason we choose a Gaussian form of the regularization is because it is the simplest example of a radially symmetric function with the same functional form in the Fourier space. Any other form of radially symmetric smooth function would have worked equally well, but we need not attempt to find such regularization because the theoretical flow obtained from the Gaussian spread to the force agrees very well with our experimentally observed one.

d) Both the data and the model are for time-averaged flow field, which loses the (large) oscillations occurring in the near-field flow due to the flagellar beat. These oscillations may have a role in mixing. Resolving this flow is probably very difficult experimentally, but can be accomplished in silico through computational modelling. The difference between time-averaged and instantaneous flow should at least be acknowledged.

We thank the reviewer for raising this important point. We completely agree with the reviewer that flow oscillations can indeed play a role in fluid advection and mixing, as also suggested by the reviewing editor. An order of magnitude estimates of flagella-driven flow oscillations (vosc) suggest that advection is important for both the H30 (νb∼55 Hz, vosc∼3450 μm/s) and H10 (νb∼52 Hz, vosc∼3270 μm/s) cells as their beat frequencies (νb) are similar. However, in this study, we try to understand the long-time behaviour of the flows where the oscillations are averaged out and we observe that even these time-averaged flows of swimming CR cells have interesting flow structures and transport properties when strongly confined. Nevertheless, we understand and appreciate the spirit of the concern raised by the reviewer and acknowledge it by including the following sentences in the discussion of our revised manuscript (page 21).

“We note that apart from the time-averaged flows, the oscillations produced in the flow (vosc) due to the periodic beating of the flagella can play a role in fluid transport and mixing for both the H30 (νb∼55 Hz, order of magnitude estimate of vosc∼L×2πνb∼3450 μm/s) and H10 (νb∼52 Hz, vosc∼3270 μm/s) cells [Guasto et al., 2010; Klindt and Friedrich, 2015].”

e) The slower decay of the velocity correlation in the confined case (figure 5 – supplement figure 2) could be explained by the fact that the confined case produces a force monopole, and hence a lower order of decay than a force-free swimmer? In which case we are not necessarily seeing evidence of increased mixing?

We thank the reviewer for raising this pertinent concern. We agree with the reviewer on this point that the slower decay of velocity correlation (revised *Figure 5A*) in the confined case can be ascribed to lower order of decay in the swimmer’s flow field. This is because strong confinement reduces the force-free swimmer in H30 (weakly confined force-dipole with 1r3 decay) to a force-monopole one in H10 (1r2 decay). However, we respectfully differ with the concluding remark of the reviewer that a slower decay rate in fluid flow does not necessarily imply increased mixing. Rather, it can be one of the reasons contributing to increased mixing in microswimmer suspensions. That is, a velocity field with RMS value V and correlation length λ (equivalently, persistence time τ=λ/V) will contribute an amount Vλ=V2τ  to the diffusivity, on time scales ≫τ. Thus, the more persistent the velocity, the larger the diffusivity enhancement of the fluid particles on long timescales. For example, Kurtudulu et al., 2011 observe enhanced mixing in active CR suspensions in freely-suspended 2D soap films compared to those in 3D unconfined fluid [Leptos et al., 2009]. They observe higher effective diffusivity of passive tracers in 2D and attribute this increase to the long-range hydrodynamic disturbances of the swimmers due to the reduced spatial dimension in their experiments (the force-dipolar flow reduces from v∼1r2 in 3D to v∼1r in 2D) and also due to increased swimmer-tracer interactions.

Nevertheless, we understand the spirit of the concern raised by the reviewer and acknowledge it by supporting our claim of enhanced mixing with an additional analysis, in a manner similar to those of [Kurtudulu et al., 2011], which directly shows that diffusivity of passive tracers increases due to the strongly confined (yet slow-swimming) CR’s flow field. We measure the displacement of the passive tracer particles (200 nm microspheres) when a single swimmer passes through the field of view (179 μm × 143 μm) in our experiments. The H30 swimmers are fast and therefore pass through these field of view in ~ 1-1.4 sec (revised *Figure 5*B) whereas the slow-moving H10 swimmers stay in the field of view for the maximum recording time of ~ 6-8 sec (revised *Figure 5*C). As the swimmer moves within the chamber, it perturbs the tracer particles. The trajectories of these tracer particles involve both Brownian components and large jumps induced by the motion and flow field of these swimmers. We colour code the tracer trajectories based on their maximum displacement, Δr, during a fixed lag time of 0.2 second (∼10 flagellar beat cycles) (revised *Figure 5B,C*). The tracer trajectories close to the swimming path of the representative H30 swimmer (black dashed arrow) are mostly advected by the flow whereas those far away from the cell involve mostly Brownian components (revised *Figure 5*B). However, most of tracers in the full field of view are perturbed/advected due to the H10 flow, those in the close vicinity being mostly affected (revised *Figure 5*C). Their advective displacements are larger than that of the tracers due to H30 flow. Together both these representative images (revised *Figure 5B,C*) show that the spatial range to which a swimmer motion advects the tracers – radius of influence, Rad – is higher in the case of H10 flow (Rad≈35 μm) when compared to the H30 one (Rad≈15 μm). We define the radius Rad to be approximately equal to the lateral distance from the cell’s swimming path (black dashed arrow) where the tracer displacements decrease to ∼ 20% of their maxima (dark orange trajectories). The region of influence for the H30 cell is a cylinder of radius Rad≈15 μm with the cell’s swimming path as its axis and that for the H10 cell is a sphere of radius Rad≈35 μm centred on the slow swimming cell’s trajectory.

We also measure the mean-squared displacement (MSD) of the tracers to quantify the relative increment in the advective transport of the H10 flow with respect to the H30 one. We calculate the MSD of approximately 500 tracers in the whole field of view for each video where a single cell is swimming through the field of view and then ensemble average over 6 such videos (revised *Figure 5 — figure supplement 1*). These plots with a scaling ⟨Δr2(Δt)⟩∝Δtα show a higher MSD exponent in H10 (α∼1.55) than H30 (α∼1.25), indicating enhanced anomalous diffusion in strong confinement.

To summarize, these plots directly show that fluid is indeed advected more in the strongly confined case (H10) than the weakly confined one (H30) leading to enhanced mixing and transport.

We acknowledge the reviewer’s concern by re-writing the section “Enhancement of fluid *mixing* in strong confinement” with these additional plots and analyses (revised Figure 5 and Figure5—figure supplement 1).

f) Apart from a brief reference to a review paper, relatively little contact is made with Chlamydomonas biology or ecology, particularly concerning the functional importance of fluid mixing. I am not a specialist in this area; my question is, under what circumstances is the ability to exchange e.g. carbon dioxide with the surrounding fluid a limiting factor in metabolism? Are we seeing something specific to CR's adaptation to certain natural habitats, or just that tethered swimmers always disturb more fluid than free swimmers due to the force monopole?

We thank the reviewer for this opportunity to make clarifications. Algal growth and metabolism require exchange, between organisms and water, of small molecules and ions such as phosphate, carbon dioxide, nitrogen etc. Specifically, nitrogen and carbon are limiting macronutrients to algal growth [Short et al., 2006; Khan et al., 2018]. For example, dissolved carbon dioxide in the surrounding fluid contains the carbon source essential for photosynthesis. Carbon dioxide further buffers the water against pH changes as a result of CO_2_/HCO_3_^-^ balance, which helps in maintaining the pH between 7 and 9, optimum for algal growth [Khan et al., 2018]. Therefore, fluid mixing helps in uniform distribution of nutrients, air and CO_2_ in algal cultures which have a positive influence on the nutrient uptake of these organisms, especially for the strongly confined cells as they cannot move far enough to outrun diffusion of nutrient molecules because of slow swimming speed.

However, we note that this increased fluid mixing helping the organism to avail itself of more nutrients is NOT an adaptation of the organism for being in strongly confined spaces. The inverse vortical flow field is a purely physical effect due to mechanical interaction of the cell body with the solid walls and not due to any behavioural change of CR in confined space, as mentioned in multiple places in our manuscript. Even though the walls restrict the cell body from swimming freely, they hardly affect the flagellar motion as the flagella are slender rods of diameter 0.5 μm beating far from these solid boundaries (∼ 5 μm) in the 10 μm chamber. As a result, the flagellar waveform and beat frequency in strong confinement are similar to those in the bulk. This implies that the strongly confined CR flow, that is mostly ascribed to the flagellar motion, and consequently the enhancement in fluid transport appears not to be an adaptation of the organism but a mechanical effect of being in strongly confined spaces. On the other hand, we agree with the reviewer that our observations are in line with those of tethered filter feeders like *Vorticella* whose ciliary beating produces a vortical flow field (similar to ours) that draws in fluid with dispersed bacteria towards the organism. These organisms being tethered to a substrate exert a net force towards themselves like our force-monopole swimmer in strong confinement and therefore, the flow field decays slowly than a free-swimming one [Christensen-Dalsgaard and Fenchel, 2003; Pepper et al., 2010].

We acknowledge the reviewer’s concern by modifying and adding text at appropriate places in our revised manuscript (pgs. 16 and 18 and 21).

Reviewer #2:The manuscript focusses on changes in the flow fields generated by swimming microorganisms as a consequence of them being squeezed within a gap that is narrower than their size. This is studied here in the context of the unicellular green microalga *Chlamydomonas reinhardtii*, which is a common model system for microbial motility of body sizes ranging from 8um to 14um. The authors compare their behaviour in the two cases of a 30um and 10um-thick Hele-Shaw cell. The resulting flow fields are then compared with those obtained by a superposition of quasi-2D Stokeslets, Green functions of the Brinkman equation.The paper has three main results. Firstly, cell behaviour in the 10um-thick samples depends clearly on microorganismal size. Larger cells beat their flagella with the standard synchronous breaststroke, while smaller ones display either asynchronous beating or "paddling" (as the authors call it, but see below my comment on this). Secondly, the larger cells display an average flow field that, in the far field, is directed oppositely to what would be predicted in absence of cell squeezing by the walls. Thirdly, the paper presents a theoretical modelling for the observed flow fields in terms of a "Gaussian" Brinkman Stokeslets. This flow field is proposed to increase the flow of nutrients to the cell.1) The confinement that the authors focus on is in a different regime that what has been addressed earlier, in the sense that it starts to probe strong non-hydrodynamic friction, which is a case that can definitely happen in nature. This is interesting, although likely to depend quantitatively very much on topographical and chemical details of the actual surfaces that are in close contact.Given that the strongly confined cells experience a non-hydrodynamic friction that is large enough to almost halt their swimming, it is not surprising that the measured flow field is dominated by the forces imposed on the fluid by the flagella and therefore a force-free 3-Stokeslet model is inappropriate. After all, the presence of non-hydrodynamic friction means that the cell exerts a net force on the fluid. It seems to me that this situation essentially resembles that of a tethered microorganism like Vorticella and I would have liked to see more discussion on the similarities between the two cases, both experimentally and from a modelling perspective. I do not think this is developed sufficiently in the manuscript.

We thank the reviewer for this relevant suggestion. We agree that the strongly confined cell resembles the case of a sessile filter feeder like *Vorticella* which is generally tethered to a substrate via its stalk. Below, we outline the similarities between these two cases from both experimental and theoretical perspectives.

The ciliary array of tethered filter feeders like *Vorticella* and *Stentor,* distributed along the periphery on the top of the bell-shaped body, beat at 30-50 Hz in metachronal coordination to generate a fluid flow towards themselves and feed from the passing fluid [Ryu et al., 2016]. Experimental measurements of the flow fields in the standard slide-coverslip setup with the tethered organism oriented parallelly and midway between the two surfaces show a dual-vortex flow structure [Nagai et al., 2009; Pepper et al., 2010; Ryu et al., 2016], similar to our strongly confined cell (Figure 3C). This vortical flow field generated by the ciliary carpet of *Vorticella* is much stronger than that of our strongly confined CR (2-cilia flow) with a maximum velocity of 360 μm/s drawing in fluid containing food particles about 450 μm from the body of the organism [Nagai et al., 2009; Ryu et al., 2016].

It is evident from these experimental flows that the tethered feeders exert a net force towards themselves which is modelled by a parallel Stokeslet between two rigid walls, directed towards the body of the organism, which explains the far-field flow features reasonably well [Pepper et al., 2010; Ryu et al., 2016]. This Stokeslet model can also explain our strongly confined CR’s flow-field, far from the organism, because the CR squeezed between the two walls is barely able to move and hence the flagella pull fluid from front to back resulting in a net force on the fluid towards the cell. However, we show that the near-field flow due to the two flagellar motion of the strongly confined CR is accurately described by two (like-signed, directed towards the cell body) Brinkman Stokeslets localized with a Gaussian spread on the approximate flagellar positions. Similarly, Pepper et al., considered an effective 2D Brinkman cylindrical model to account for finite-size effects, where the *Vorticella* is modelled as a cylinder (axis perpendicular to the walls) with a tangential velocity distribution on the surface to describe the multiciliary beating on the top of the filter feeder [Pepper et al., 2010]. This analytical model is slightly more involved than ours due to the coordinated beating of multiple cilia for filter feeders. To summarize, these Stokeslet and Brinkman flow models agree well with the experimentally observed flow vortices of *Vorticella* and strongly confined CR with appropriate consideration of the differences in their ciliary beating (multi-ciliated metachronal waves for Vorticella and two-ciliary flow for CR).

We acknowledge the reviewer’s suggestion by adding a short summary of the above in the discussion of our revised manuscript (page 21) in order to keep the storyline focussed.

Besides this, the range of gaps probed by the authors is rather coarse. The 10um gap is not much smaller than the smallest gaps that were probed in ref [12] for which -however- the flow field was qualitatively different. In my opinion, within the context of the effect of confinement on flow fields, it would have been interesting to explore this range in more detail, studying the transition from the hydrodynamic to the contact-friction case.

We thank the reviewer for raising this pertinent suggestion. It is a question that we ourselves have wondered about but can think of no simple experimental or theoretical approach to answer. We cannot produce double tape spacer with the necessary resolution. In principle, one can imagine designing an experiment where the cell and tracer suspension is injected into a microfluidic chamber made of glass walls and PDMS spacer, and the PDMS is pumped to gradually increase the chamber height while simultaneously observing the changes in the CR flow field. This can be perhaps explored in a future study. However, even in the 10 μm chamber, cells which are less confined (H10 Wobblers, D/H ≈ 1) have an irregular beat pattern due to frequent interactions of flagella with the rigid walls of the chamber during the spinning motion of the cells and therefore, it is difficult to assign a beat-averaged flow field that is universal for all H10 Wobblers. So, it is challenging to measure an ensemble averaged flow field with gradually decreasing confinement, especially when one attempts to observe the transition from contact friction case (our strongly confined cell, D/H ≈ 1.2, force-monopole flow, akin to a 2D source-dipolar flow pointing *opposite* to swimmer’s motion) to the hydrodynamic one (Jeanneret et al., 2019, D/H ≈ 0.7, force-free 2D-source-dipolar flow *along* the swimmer’s motion). It is quite clear that this transition is discontinuous, where individual cells in the chamber will be probing either of these two cases even when their sizes may be similar.

As for the theoretical approach to studying the transition between the two limits, we would need to build a model that accounts for binding and unbinding of the molecular entities on the cell surface with the walls. Such a model will of course involve several phenomenological parameters that would act as additional degrees of freedom while comparing experimental data to the theoretical predictions. Furthermore, such an effort would also demand a more controlled experimental system wherein the surface chemistry can be tuned to vary the frictional interactions.

2) Regarding the modelling, I have two main comments. Firstly, I appreciate the idea of a diffused Stokeslet. However, it does look very close to the idea of a regularised Stokeslet (Cortez 2005), which is not even mentioned in the manuscript. I think this is quite surprising. I would expect the manuscript to comment on differences between the two approaches.

We thank the reviewer for appreciating our idea of a diffused Stokeslet and seeking comparison of our approach with that of Cortez et al., 2005. Cortez have introduced the idea of a regularized Stokeslet by including a radially symmetric smooth function ϕϵ (e.g., Gaussian, Lorentzian etc.) in the forcing term to stabilize the singularities in the Stokeslet expression for practical computation [Cortez, 2001]. This is done by introducing a small cut-off parameter ϵ, in the function ϕϵ, which controls the spreading. Cortez et al., have provided a methodology to compute the closed-form analytical solutions for any such functional regularization ϕϵ to the Stokeslet [Cortez, 2001; Cortez et al., 2005]. Our approach of including a Gaussian form of regularization to the Brinkman Stokeslet is in a similar spirit. We have convolved our Brinkman Stokeslet to a 2D Gaussian with standard deviation σ (which is similar to ϵ of Cortez et al.,) to numerically obtain the flow-field. That is, our approach is only different from Cortez et al., in its implementation. We have given numerical solution to the regularized Stokeslet so that it explains our experimental data while Cortez et al., have provided analytical expressions for a Lorentzian regularization in an unbounded domain [Cortez, 2001; Cortez et al., 2005]. To summarize, our convolution approach gives numerical solution to any form of regularization to the Stokeslet whereas Cortez et al., approach gives analytical closed-form solutions to the same.

We understand the spirit of the concern raised by the reviewer and acknowledge it by elaborating the following sentence while introducing the Gaussian regularization in our revised manuscript (page 15):

“We, therefore, associate a 2D Gaussian source of standard deviation σ, to Equation 1 instead of the point-source δ(r), in a manner similar to the regularized Stokeslet approach [cite Cortez et al., 2005]”

Secondly, I do not understand the FT approach to finding the 2D Stokeslet for the Brinkman equation, when this is actually known analytically. It has been published inPushkin, D. O. and Bees, M. A. Bugs on a slippery plane: Understanding the motility of microbial pathogens with mathematical modelling. Adv. Exp. Med. Biol. 915, 193-205 (2016).which is cited in ref [12] of the manuscript (Jeanneret et al., PRL 2019). This paper also shows that a model based on the point forces from Liron and Mochon does not perform well (see Suppl Mat), while one made of Brinkman Stokeslets with a 2D source dipole does. Therefore I do not think that it is surprising to see that the same approach works well for the current manuscript. In fact, given the similarity between the current setup and that in [12], I would have expected a comparison between the current approach and the full model from [12] at least in the Supplementary Material.

We appreciate the reviewer for seeking comparison of our theoretical model with that of Jeanneret et al., 2019 [12] which uses the Pushkin-Bees (PB) solution [Pushkin and Bees, 2016] in constructing their 2D model.

First, we compare our Brinkman equation with that of Pushkin and Bees, 2016 for the 2D fluid velocity v(x,y). Eq 12.2 of [Pushkin and Bees, 2016] for the 2D fluid velocity, averaged over the film thickness of height H in the z-direction, is given by the following,

−∇p(x,y)+η∇2v(x,y)−ηλ2v(x,y)=0

Pushkin and Bees considered the permeability length to be λ=H for a channel that is formed by 2 parallel solid plates with no-slip boundary conditions i.e., a Hele-Shaw cell [Pushkin and Bees, 2016]. Jeanneret et al., corrected this factor to be λ=H12 for the *z-averaged mean 2D Poiseuille flow* in a Hele-Shaw cell, relevant for the comparison with their experimental flow field which is averaged over the depth of focus of ∼10 μm (20X objective; numerical aperture, NA, 0.4) [Jeanneret et al., 2019]. On the other hand, we write the quasi-2D Brinkman equation at the z=0 plane (Equation 1 and associated text in our manuscript) for appropriate comparison with our experimental data which is acquired at or nearby the mid-plane (z=0 between the solid walls at z=±H2) with a 40X objective that has very low depth of focus (∼1 μm). The permeability length of our Equation 1, on comparing with the above one, is λ=Hπ. Therefore, we use the exact analytical solution given by Pushkin and Bees in Equation 12.6-12.8 [Pushkin and Bees, 2016] with λ=Hπ for comparison with our Brinkman Stokeslet’s numerical solution using the Fourier Transform (FT) method. We find that the flow field obtained for 2-Stokeslets using the analytical solution (PB) is identical to that of our numerical one (revised Figure 4A in our manuscript) computed on the same grid size and spacing. Hence, our FT methodology for solving the Brinkman equation *from first principles* is an alternative approach to using the complete analytical solution of Pushkin and Bees. We appreciate the reviewer’s concern and acknowledge it by including the following sentence in our revised manuscript below Equation 2:

“This solution is identical to the analytical closed-form expression of Pushkin and Bees, 2016.”

We also agree with the reviewer that Jeanneret et al., 2019 [12] have shown that their theoretical model including a force-free combination of Pushkin-Bees (Brinkman) Stokeslets along with a 2D source dipole performs well over the spinning force-free combination of Liron and Mochon (LM) Stokeslets. However, our approach is not exactly the same. We show that force-free Brinkman Stokeslets (Figure 3 —figure supplement 1A) *does not* explain our experimentally observed flow in H=10 μm (Figure 3C) even qualitatively. On the other hand, we show that a quasi-2D Brinkman Stokeslet/force-monopole with a Gaussian regularization (spatially distributed between 2 flagellar positions; revised Figure 4C) matches our experimental flow field very well. Jeanneret et al., 2019 added the extra 2D source dipole because they observed that the spinning force-free combination of LM Stokeslets lacks the dipolar symmetry due to finite-sized body effects when compared with the experiments (Figure S6 of [Jeanneret et al., 2019]). However, when we subtract the theoretical 2-Stokeslet LM flow (not force-free; see Appendix1 – figure 1A) from our H10 experimental one (Figure 3C), we observe that we need spatially-averaged or regularized forces instead of point forces to explain our experiment.

The reason why our theoretical approach is not the same as Jeanneret et al., is because there are two major differences in our experimental observations:

a. In our case, the strongly confined CR exerts a net force on the fluid due to thepresence of strong non-hydrodynamic contact friction from the walls, unlike that of [Jeanneret et al., 2019]. This leads to a 96% reduction of the swimming speed of the CR cells when strongly confined in 10 μm chamber (D/H ~ 1.2) as compared to 30 μm chamber (D/H ~ 0.3) in our experiments. This coupling between motility and confinement is not observed by Jeanneret et al., likely due to the slightly weak confinement (D/H ~ 0.7) produced by their experimental methodology (Table S1 of [Jeanneret et al., 2019]), where the stresses present in the system are mostly hydrodynamic. It is therefore appropriate for them to use the conventional 3-Stokeslet theoretical model for CR which is force-free (apart from the source dipole contribution) whereas in our case, the nearly absent hydrodynamic drag experienced by the cell body leads to a monopolar flow with only 2 Stokeslets (like-signed) localized with a Gaussian spread around the approximate flagellar positions.

b. The CR cells in our experiment *do not spin* around their body axis in the strong confinement of H=10 μm, contrary to the experimental observation of Jeanneret et al. They added the extra 2D source dipole in their theoretical model to the force-free Pushkin-Bees Stokeslets to account for both finite-sized effects of the cell body and spinning motion of the cells (explained in Figure 1(c) of [Jeanneret et al., 2019]).

To further illustrate our point and appreciating the suggestion of the reviewer, we provide a detailed comparison of our theoretical model with that of Jeanneret et al., 2019 [12] in a new section in the Appendix of our revised manuscript – ‘5. Comparison of our theoretical model of strongly confined flow with that of Jeanneret et al.’.

Finally, I think that comparing the magnitude of the flow fields as in Figure 5B,D is insufficient. One should instead show that both magnitude and direction of the flow fields are well captured by the model. This cannot really be grasped by comparing by eye Figure 3C and Figure 5A, C.

We thank the reviewer for raising this concern. We have already given the root mean square deviation (RMSD) between the experimental and theoretical flows in vx, vy and |v|, calculated at all grid points, for a quantitative comparison for the direction as well as the magnitude of flow fields. To further address the reviewer’s concern, we now show the comparison of the x and y components of the velocity field (vx and vy), whose magnitudes are responsible for determining the direction of the flow. We add another figure in the revised manuscript (revised Figure4 —figure supplement 2) which shows the azimuthal variation of vx and vy for representative radial distances of the H10 experimental flow fields and the corresponding theoretical models. We also add the corresponding comparison between H30 experiment and its theory in subfigures C and D of the revised Figure4 —figure supplement 3.

We do not add the comparison in the velocity components between H10 experiment and Liron-Mochon theory as this model is not appropriate for describing the experiment, as expected, and described in the manuscript. This deviation is already captured while comparing their flow magnitudes (Appendix1-Figure1) and hence we do not expect their directions to match either.

3) As for the question of nutrient transport, I find the claim on the enhancement for the 10um-cells not completely convincing. It is clear that the fluid fluxes for the 30um-cells and the 10um-cells are organised in an opposite way. However, Figure 4D shows that the positive part of the flux is of a very similar magnitude in both cases, in particular for the average fluxes. This is true also when one compares the negative fluxes. Given the similarities, I do not find a compelling reason why the cell should have a much higher nutrient uptake in one of the two cases. This might be the case, but there is not enough evidence in the paper to support this statement.

We appreciate the point raised by the reviewer. We understand that comparison of the average fluid fluxes may not be completely convincing to our claim of increased nutrient availability to the organism when strongly confined. We now support our claim of enhanced mixing and thereby nutrient transport through the fluid with an additional analysis which directly shows that diffusivity of passive tracers increases due to the strongly confined (yet slow-swimming) CR’s flow field. We measure the displacement and the mean-squared displacement (MSD) of the passive tracer particles (200 nm microspheres) when a single swimmer passes through the field of view (179 μm × 143 μm) in our experiments [Kurtudulu et al., 2011, Leptos et al., 2009]. Please refer to revised Figure 5 and revised Figure 5 — figure supplement 1, where we plot these quantities.

Nutrient uptake by any organism will depend on how much nutrient molecules the flow is bringing to the organism and the spatial scale/structure of the flows (apart from the absorption by the cell). Below, we summarize the results from our previous as well as new analyses which corroborate our statement of enhanced fluid transport/mixing for the 10 μm cells which have a positive influence on the nutrient uptake of these cells as they cannot move far enough to outrun diffusion of nutrient molecules because of slow swimming speed.

i. The slower decay in the fluid velocity correlation in the strongly confined case (revised Figure 5A) shows that the characteristic structure and spatial scale of the H10 flow is larger than the H30 one. This leads to longer correlation length scales in the flow velocity, which implies an increased effective diffusivity (scaling, ∼Vrmsλ for a velocity field with RMS value Vrms and correlation length λ) of the fluid particles on time scales ≫λ/Vrms, in strong confinement.

ii. *Figure 5, B and C* showing colour coded tracer trajectories (according to their maximum displacement within a fixed Δt=0.2s) due to the motion and flow field of a representative H30 and H10 cell through the field of view also supports the above point as follows. The tracer trajectories close to the swimming path of the H30 swimmer (black dashed arrow) are mostly advected by the flow whereas those far away from the cell involve mostly Brownian components (revised Figure 5-A). However, most of tracers in the full field of view are perturbed/advected due to the H10 flow, those in the close vicinity being mostly affected (revised Figure 5-B). Their advective displacements are larger than that of the tracers due to H30 flow. Together both these representative images (revised *Figure 5*) show that the spatial range to which a swimmer motion advects the tracers – radius of influence, Rad – is higher in the case of H10 flow (Rad≈35 μm) when compared to the H30 one (Rad≈15 μm).

iii. Revised Figure 5 — figure supplement 1 shows the MSD of approximately 500 tracers in the whole field of view for each video where a single cell is swimming through the field of view and then ensemble average over 6 such videos. These plots with a scaling ⟨Δr2(Δt)⟩∝Δtα show a higher MSD exponent in H10 (α∼1.55) than H30 (α∼1.25) indicating enhanced anomalous diffusion in strong confinement. Therefore, MSD calculation of the tracers quantify the relative increment in the advective transport of the H10 flow with respect to the H30 one.

We understand the spirit of the concern raised by the reviewer and acknowledge it by removing the volume flux calculation and subfigures from our revised manuscript. Instead, we rephrase and modify the concerned section: “Enhancement of fluid *mixing* in strong confinement” with these new analyses (please see revised Figure 5 and Figure 5—figure supplement 1) that directly address the queries of the reviewer.

4) Finally, regarding the "paddling" state, I'm afraid this is the normal flagellar shock response in Chlamydomonas and not some sort of previously unknown state of the cells. It is known that the shock response can be elicited both by intense light stimuli or by mechanical stimuli. It is very likely that this is just a mechanosensitive shock, which comes from the mechanical interactions between the flagella and the upper/lower walls. In turn, these are possible due to the fact that cells of smaller size than the gap can spin around their body enough to touch the walls with their flagella. I think that the authors would need to investigate further the existing literature on flagellar shock response in Chlamydomonas and put appropriately in context the "paddling" behaviour they observe.

We thank the reviewer for raising this important concern. Mechanosensitive ion channels in CR, activated by mechanical agitation and/or compressive stresses, induce an influx of calcium ions which are responsible for changes in flagellar waveform (from breastroke to undulatory) leading to transient backward motion of the cell [Yoshimura et al., 1997; Fujiu et al., 2011] and/or increased beat frequency [Wakabayashi et al., 2009]. They also affect the inter-flagellar coordination through calcium-sensitive basal-body associated fibrous structures [Ruffer and Nultsch, 1998]. We agree with the reviewer that these calcium-mediated mechanosensitive shock responses to the flagellar beating in CR are very likely the reason for the ‘paddling’ behaviour observed in our experiments due to frequent flagella-wall interactions during the spinning motion of the cells when they are slightly smaller than the gap between these walls. We acknowledge the reviewer’s suggestion by adding/modifying texts in our revised manuscript (pg. 7-8) related to the paddler flagellar beat.

References:

[Drescher et al., 2010] K. Drescher, R. E. Goldstein, N. Michel, M. Polin, I. Tuval, Direct measurement of the flow field around swimming microorganisms. *Phys. Rev. Lett.* 105, 168101 (2010).

[Ishikawa et al., 2006] T. Ishikawa, M. P. Simmonds, T.J. Pedley, Hydrodynamic interaction of two swimming model micro-organisms. *J. Fluid Mech.* 568, 119-160 (2006).

[Mathijssen et al., 2015] A. J. T. M. Mathijssen, D. O. Pushkin, J. M. Yeomans, Tracer trajectories and displacement due to a micro-swimmer near a surface. *J. Fluid Mech.* 773, 498-519 (2015).

[Jeanneret et al., 2019] R. Jeanneret, D. O. Pushkin, M. Polin, Confinement enhances the diversity of microbial flow fields. *Phys. Rev. Lett.* 123, 248102, (2019).

[Pushkin and Bees, 2016] D. O. Pushkin, M.A. Bees, Bugs on a slippery plane: Understanding the motility of microbial pathogens with mathematical modelling. *Adv. Exp. Med. Biol.* 915, 193-205 (2016).

[Cortez, 2001] R. Cortez, The method of regularized Stokeslets. *SIAM J. Sci. Comput.* 23, 1204-1225 (2001)

[Cortez et al., 2005] R. Cortez, L. Fauci, A. Medovikov, The method of regularized Stokeslets in three dimensions: Analysis, validation, and application to helical swimming. *Physics of Fluids* 17, 031504 (2005)

[Guasto et al., 2010] J. S. Guasto, K. A. Johnson, J. P. Gollub, Oscillatory Flows Induced by Microorganisms Swimming in Two Dimensions. *Phys. Rev. Lett.* 105, 168102 (2010).

[Klindt and Friedrich, 2015] G. S. Klindt and B. M. Friedrich, Flagellar swimmers oscillate between pusher- and puller-type swimming. *Phys. Rev. E* 92, 063019 (2015).

[Wu and Libchaber, 2000] X.-L. Wu, A. Libchaber, Particle Diffusion in a Quasi-Two-Dimensional Bacterial Bath. *Phys. Rev. Lett.* 84, 3017-3020 (2000).

[Liron and Mochon, 1976] N. Liron and S. Mochon. Stokes flow for a stokeslet between two parallel flat plates. *Journal of Engineering Mathematics* 10, 287-303 (1976).

[Mathijssen et al., 2016] A. J. T. M. Mathijssen, A. Doostmohammadi, J. M. Yeomans, T. N. Shendruk. Hydrodynamics of micro-swimmers in films. *Journal of Fluid Mechanics* 806, 35-70 (2016).

[Brotto at al., 2013] T. Brotto, J.-B. Caussin, E. Lauga, D. Bartolo. Hydrodynamics of confined active fluids. *Phys. Rev. Lett.* 110, 038101 (2013).

[Kurtudulu et al., 2011] H. Kurtuldu, J. S. Guasto, K. A. Johnson, J. P. Gollub, Enhancement of biomixing by swimming algal cells in two-dimensional films. *Proceedings of the National Academy of Sciences* 108, 10391– 10395 (2011).

[Leptos et al., 2009] K. C. Leptos, J. S. Guasto, J. P. Gollub, A. I. Pesci, R. E. Goldstein, Dynamics of Enhanced Tracer Diffusion in Suspensions of Swimming Eukaryotic Microorganisms. *Phys. Rev. Lett.* 101, 198103 (2009).

[Khan et al., 2018] M. I. Khan, J. H. Shin, J. D. Kim. The promising future of microalgae: current status, challenges, and optimization of a sustainable and renewable industry for biofuels, feed, and other products. *Microbial Cell Factories* 17, 36–36 (2018).

[Short et al., 2006] M. B. Short, C. A. Solari, S. Ganguly, T. R. Powers, J. O. Kessler, R. E. Goldstein, Flows driven by flagella of multicellular organisms enhance long-range molecular transport. *Proceedings of the National Academy of Sciences* 103, 8315 – 8319 (2006).

[Christensen-Dalsgaard and Fenchel, 2003] K. K. Christensen-Dalsgaard, T. Fenchel, Increased filtration efficiency of attached compared to free-swimming flagellates, *Aquat. Microb. Ecol.* 33, 77 – 86 (2003).

[Pepper et al., 2010] R. E. Pepper, M. Roper, S. Ryu, P. Matsudaira, H. A. Stone, Nearby boundaries create eddies near microscopic filter feeders. *J. R. Soc. Interface* 7, 851–862 (2010).

[Ryu et al., 2016] S. Ryu, R. E. Pepper, M. Nagai, D. C. France, Vorticella: A Protozoan for Bio-Inspired Engineering. *Micromachines* 8, 4 (2016).

[Nagai et al., 2009] M. Nagai, M. Oishi, M. Oshima, H. Asai, H. Fujita, Three-dimensional two-component velocity measurement of the flow field induced by the *Vorticella picta* microorganism using a confocal microparticle image velocimetry technique. *Biomicrofluidics* 3, 014105 (2009).

[Yoshimura et al., 1997] K. Yoshimura, C Shingyoji, K Takahashi. Conversion of beating mode in *Chlamydomonas* flagella induced by electric stimulation. *Cell Motil. Cytoskeleton* 36, 236–45 (1997).

[Fujiu et al., 2011] K. Fujiu, Y. Nakayama, H. Iida, M. Sokabe, K. Yoshimura, Mechanoreception in motile flagella of *Chlamydomonas*. *Nat. Cell Biol.* 13, 630–632 (2011).

[Wakabayashi et al., 2009] K.-i. Wakabayashi, T. Ide, R. Kamiya, Calcium-dependent flagellar motility activation in *Chlamydomonas reinhardtii* in response to mechanical agitation. *Cell Motil. Cytoskeleton* 66, 736-742 (2009).

[Ruffer and Nultsch, 1998] U. Rüffer, W. Nultsch, Flagellar coordination in *Chlamydomonas* cells held on micropipettes. *Cell Motil. Cytoskeleton* 41, 297-307 (1998).